# Formulation and In Vitro Characterization of Granulocyte-Colony-Stimulating-Factor-Loaded Liposomes: Future Prospective in Reproductive Medicine

**DOI:** 10.3390/ijms26062689

**Published:** 2025-03-17

**Authors:** Kseniia V. Obedkova, Vladislava V. Khalenko, Dmitry V. Tovpeko, Julian R. Ryzhov, Olesya N. Bespalova, Natalya I. Tapilskaya

**Affiliations:** D.O. Ott Research Institute of Obstetrics, Gynecology and Reproductive Medicine, 199034 St. Petersburg, Russia; obedkova_ks@mail.ru (K.V.O.); tovpeko.dmitry@gmail.com (D.V.T.); julian.ryzhov@gmail.com (J.R.R.); shiggerra@mail.ru (O.N.B.); tapnatalia@yandex.ru (N.I.T.)

**Keywords:** infertility, granulocyte colony-stimulating factor, liposomes, liposomal drug formulations, nanoparticles, mitochondrial respiration

## Abstract

Drug delivery systems enhance drug efficacy while minimizing side effects. Liposomes, as well-studied and clinically approved carriers, offer biodegradability, biocompatibility, and low toxicity, making them suitable for delivering various pharmacological agents. Granulocyte colony-stimulating factor (G-CSF), a key growth factor, has shown therapeutic potential, particularly in infertility treatment. It effectively manages chronic and refractory endometritis by improving endometrial receptivity and increasing embryo implantation success. Studies indicate that G-CSF promotes endometrial growth and enhances the uterine microenvironment, benefiting patients with recurrent implantation failures and chronic endometritis. Encapsulation of G-CSF in liposomes enhances its stability, bioavailability, and controlled release. G-CSF-loaded liposomes were prepared using passive loading via the thin-film hydration method. The size of the liposomes, polydispersity index (PDI), and zeta potential were determined using dynamic and electrophoretic light scattering methods, and the encapsulation efficiency was measured using high-performance liquid chromatography. The morphology of the liposomes was established and confirmed using cryogenic transmission electron microscopy. The cytocompatibility of the G-CSF-loaded liposomes was evaluated on human dermal fibroblasts using an MTT assay. The G-CSF-loaded liposomes had an average particle size of 161.9 ± 9.9 nm, a PDI of 0.261 ± 0.03, and a zeta potential of +2.09 ± 0.10 mV, exhibiting high physical stability during long-term storage at +4 °C and 60% humidity. The passive loading method resulted in a 52.37 ± 3.64% encapsulation efficiency of the active substance. The analysis of cell viability revealed no cytotoxicity toward liposomes loaded with G-CSF and demonstrated a dose-dependent effect on the viability of human dermal fibroblasts. Thus, the obtained data confirm the successful preparation of G-CSF-loaded liposomes. However, to fully understand their effectiveness in biomedical applications, further research is needed, including an evaluation of their effectiveness in vivo. Such studies will help in determining the potential of these formulations for specific biomedical purposes and evaluating their safety and efficacy in living systems.

## 1. Introduction

Infertility remains a multifaceted challenge in reproductive medicine. A key priority in contemporary reproductive research is the development of strategies to enhance pregnancy rates in assisted reproductive technology (ART) programs and improve clinical outcomes. Despite advancements in ART, the success rates of these protocols generally remain within the range of 35–45%. One of the critical factors affecting implantation success is the presence of a thin and refractory endometrium, which continues to be an area of active investigation. Emerging therapeutic approaches, including stem cell therapy, granulocyte colony-stimulating factor (G-CSF), and peripheral blood mononuclear stem cells (PBMScs), are being explored to optimize endometrial receptivity and improve reproductive outcomes [1,2].

G-CSF has demonstrated efficacy in several studies, showing its ability to stimulate the proliferation of endometrial cells [3]. However, drug delivery remains a challenge, as conventional G-CSF formulations have a short half-life and require frequent administration. A liposomal formulation is essential as it provides sustained release of G-CSF, reducing the need for repeated administration, protecting from rapid degradation, increasing bioavailability, and reducing systemic side effects due to controlled release and targeted delivery to the endometrium. The method of administration is also crucial: local application (e.g., intrauterine delivery) may maximize G-CSF concentration in the endometrium, while systemic administration may offer prolonged effects but with a higher risk of systemic side effects.

Drug delivery systems have a high potential to increase the therapeutic index of drugs by increasing their concentration and residence time in target cells and by minimizing side effects [4,5]. Various nanostructures (dendrimers, micelles, liposomes, polymeric and inorganic nanoparticles of various nature, carbon nanotubes, etc.) are used as transport systems for the targeted delivery of active pharmaceutical ingredients to the site of action (to lesions/targets), thereby improving pharmacokinetics and drug biodistribution [5,6,7]. Of these, the most studied and approved for use in clinical practice are liposomes [8]. They are spherical lipid vesicles (usually from 50 to 500 nm in size) formed from one or more lipid bilayers by the emulsification of natural or synthetic lipids in an aqueous medium. The water volume inside the liposomes is isolated from the external environment [4,6,9]. Liposomes have a number of useful properties, such as biodegradability, biocompatibility, and low toxicity [4,6]. Due to these properties, they can be used to deliver various pharmacological agents, including proteins, peptides, and other biologically active substances [10,11].

Granulocyte colony-stimulating factor (G-CSF, *CSF3*) is a glycoprotein that performs numerous functions in both physiological and pathological processes [12,13]. G-CSF induces proliferation, the differentiation of granulocytic progenitor cells, and the maturation of precursor cells into neutrophils [12,14,15]. Natural, mature human G-CSF exists in two forms, consisting of 174 and 177 amino acids [14,16]. In clinical practice, drugs obtained by recombinant DNA technology in the *Escherichia coli* bacterial expression system (Filgrastim) or derived from Chinese hamster ovary cells (Lenograstim) are used based on a more common and more active form consisting of 174 amino acids [14,16,17]. The recombinant G-CSF molecule produced by *Escherichia coli* has an additional methionine residue at the N-terminus of the molecule and is not glycosylated compared to native G-CSF and recombinant G-CSF obtained in mammalian cells. The molecule of recombinant G-CSF contains one free cysteine residue and two disulfide bonds [14]. Due to the key role of G-CSF in the differentiation and growth of neutrophils and their precursors, it is essential for the normal functioning of the immune system. In patients suffering from diabetic foot syndrome, pneumonia, human immunodeficiency virus, febrile neutropenia, and leukemia, the use of G-CSF may help maintain immune system functions [12,18,19].

In clinical practice, the G-CSF administration in oncology patients effectively restores the level of peripheral granulocytes, which decreases as a result of high-dose chemotherapy and radiation therapy for malignant tumors, thus preventing the development of infectious complications [20,21]. G-CSF is also widely used for the mobilization of peripheral blood stem cells in autologous hematopoietic stem cell transplantation [22,23]. In the United States, G-CSF products are approved for the early treatment of radiation injuries [24]. Several clinical studies have investigated the effect of G-CSF in the treatment of thin endometrium [25,26,27,28] and its influence on oocyte maturation and embryo development in women with diminished ovarian reserve [29]. In vivo studies on a rat excision wound model demonstrated more effective wound healing in cases of local G-CSF administration in contrast to its systemic administration [30]. In a clinical study, the direct G-CSF application to the wound surface resulted in a significant healing improvement in six patients with refractory trophic ulcers [31]. G-CSF has also been successfully used to promote wound healing in patients with neutropenia [32] and dystrophic epidermolysis bullosa [33].

However, commercially available forms of G-CSF have a short pharmacological effect, while their systemic administration can cause various side effects, limiting its effectiveness [13,34,35]. The aim and novelty of the current study was to obtain G-CSF-loaded liposomes sized less than 200 nm with low polydispersity (below 0.3) and high encapsulation efficiency, providing their long-term stability. The size of liposomes was selected based on literature data that liposomes less than 200 nm interact with cell membranes while avoiding complement activation and subsequent severe toxicity [36,37]. Furthermore, the polydispersity index (PDI) was kept below 0.3, which is indicative of a uniform size distribution of liposomes.

We also conducted an initial in vitro assessment of the cytotoxicity of the developed liposomal formulations. Achieving the aforementioned characteristics of liposomes will further facilitate their application to clinical practice, ensuring increased safety and efficacy of liposomal formulations.

## 2. Results

### 2.1. Obtaining Granulocyte-Colony-Stimulating-Factor-Loaded Liposomes

In our study, G-CSF-loaded liposomes were obtained by passive loading by dispersing the lipid film in PBS containing G-CSF, followed by the sonication of multilamellar vesicles in an ultrasonic bath and membrane filtration through a sterilizing filter. Soy PC used in the preparation of G-CSF-loaded liposomes and blank liposomes was chromatographically pure and non-oxidized (oxidation index = 0.22).

The results of the particle size distribution and zeta potential measurements of the obtained liposomes are presented in Table 1. The average particle size for both blank and G-CSF-loaded liposomes was in the range of 50 to 200 nm (Figure 1), and the PDI value was less than 0.3, indicating that the samples have a highly monodisperse and uniform state [38].

All formulations were neutrally charged. It is known that neutral liposomes have a longer circulation time and lower cytotoxicity compared to cationic liposomes, as they do not interact with negatively charged proteins in the serum and blood. Additionally, neutral liposomes exhibit delayed clearance in vitro compared to cationic liposomes [39].

The obtained G-CSF-loaded liposomes were characterized by a particle size of approximately 161.9 ± 9.9 nm, a low PDI value of 0.261 ± 0.03, and a zeta potential of +2.09 ± 0.10 mV (Figure 1, Table 1). Additionally, significant improvement was achieved in encapsulation efficiency (52.37 ± 3.64%).

### 2.2. Evaluation of Granulocyte-Colony-Stimulating-Factor-Loaded Liposome Morphology by Cryogenic Electron Microscopy

Cryogenic transmission electron microscopy (cryo-EM) revealed that G-CSF-loaded liposomes appear as vesicles of regular oval, round, and, rarely, droplet shapes, ranging from 50 to 200 nm in size (Figure 2b), which confirms the results obtained using dynamic light scattering for the liposome size.

At the same time, empty liposomes consisted of unilamellar, multilamellar, and multivesicular vesicles of various sizes (Figure 2a), not exceeding the pore size of the membrane used in the final stage of liposome preparation.

### 2.3. The Physical Stability of Granulocyte-Colony-Stimulating-Factor-Loaded Liposomes During Long-Term Storage

Liposomes can be prone to physical instability, such as aggregation and fusion during storage [39], due to the influence of the external environment and other factors, causing an increase in nanoparticle size or even precipitation.

The G-CSF-loaded liposome formulation was stable for at least 2 months of storage at a temperature of 4 °C and 60% relative humidity, with no changes in particle size, PDI, and zeta potential parameters (Table 2, Figure 3). No signs of degradation were noted. The dynamic observation revealed no significant differences between all three parameters at different points in time (*p* > 0.05).

### 2.4. Assessment of the Effect of G-CSF-Loaded Liposomes on the Metabolic Activity of Human Dermal Fibroblasts

The live-cell analysis of the morphological state allowed for the assessment of the nature and extent of changes occurring in the cells under the influence of various factors (Figure 4, Figure 5, Figure 6 and Figure 7).

In the presented images (Figure 4), it can be seen that human dermal fibroblasts (DF) in the control group exhibit a characteristic MSC-like morphology and form a subconfluent monolayer.

When DF are cultured for 24 h in model media containing a G-CSF solution (Figure 5), they also form a subconfluent monolayer. However, at a G-CSF concentration of 3.2 µM, a few highly vacuolized cells (green arrows) are observed. This may indicate cellular stress or disruption of intracellular substance transport, which is often associated with exposure to high concentrations of biologically active molecules. At other G-CSF concentrations, the cells maintain normal morphology, suggesting their resilience to lower doses of this factor.

DF cultured for 24 h in model media containing G-CSF-loaded liposomes (Figure 6) and blank liposomes (Figure 7) formed a less dense monolayer compared to the control. The cells appeared compressed and thinned (blue arrows), and at concentrations of 1.6 and 3.2 µM, rounded and detached cells (red arrows) were observed.

The granular cytoplasm of the cells and the presence of artifacts (black arrows) on the surface of DF and in the surrounding medium indicate possible damage to cell membranes or degradation of the lipid components of the liposomes. The increase in the number of artifacts with rising liposome concentrations in the culture medium confirms the dose-dependent nature of this effect. Notably, in model media containing G-CSF-loaded liposomes, the artifacts floated and clustered into larger “clouds” (white arrows). This may be associated with lipid particle aggregation or their interaction with medium proteins, which intensifies with increasing concentrations of G-CSF-loaded liposomes.

Additionally, to assess the impact of liposomes on the cells, an MTT assay was conducted. This is a standard colorimetric method for analyzing metabolic activity and cell viability. The assay detects changes in the functional state of cells under the influence of tested substances by measuring the reduction of tetrazolium dye by mitochondrial enzymes of living cells (Figure 8, Table 3).

The MTT assay revealed differences in the viability of DF cultured for 24 h in model nutrient media containing G-CSF solution, G-CSF-loaded liposomes, or blank liposomes.

The viability of cells cultured in model media containing G-CSF solution at all tested concentrations was comparable to the control, indicating no significant negative effect of the factor itself on the cells. In contrast, the viability of DF cultured in media containing blank liposomes or G-CSF-loaded liposomes was lower than in the control across all tested concentrations. However, this reduction in viability did not reach a level that could be classified as a cytotoxic effect. Notably, there was a tendency for cell viability to increase with higher concentrations of G-CSF in liposomes.

Based on these findings, it can be assumed that the liposomal formulation provides a more prolonged effect compared to the G-CSF solution. This could potentially reduce the frequency of invasive procedures, improve treatment convenience, and decrease the need for frequent patient monitoring.

Additionally, when interpreting the MTT assay results, it is important to consider literature data [40], which suggests that MTT may penetrate liposomes and accumulate within them, leading to distorted viability measurements. This is supported by the results of live-cell morphological analysis, which indicate the presence of artifacts and structural changes in cells associated with liposome exposure. Thus, the observed decrease in cell viability detected by the MTT assay may be partially attributed not only to the cytotoxic effects of liposomes but also to the methodological limitations of the assay.

The obtained results suggest that the developed liposomes demonstrate a sufficient level of safety, as the reduction in cell viability does not reach cytotoxic levels and is accompanied only by moderate suppression of metabolic activity. However, further studies are necessary to optimize their application by determining the concentration that ensures a balance between effective therapeutic agent delivery and minimal impact on cells.

## 3. Discussion

In recent decades, researchers from leading countries have turned their attention toward investigating the impact of proteins and peptides in the regulation of fundamental physiological functions, including at the molecular level. Consequently, many new protein and peptide therapeutics have emerged in the pharmaceutical market. These products attract interest due to their natural origin and high efficacy [41,42]. However, one of the most significant challenges that must be addressed when utilizing such therapeutic agents in clinical settings is their rapid degradation in serum and quick elimination due to enzymatic breakdown, renal clearance, hepatic metabolism, and immunogenicity, which is attributed to their short half-life [42]. As a result, researchers are currently focusing on the development of novel advanced drug delivery systems to enhance the therapeutic index of these agents [42,43,44].

Liposomes offer several advantages over traditional pharmacologic delivery systems, especially when administered parenterally. They enhance the stability, biological activity in vivo, and therapeutic efficacy of encapsulated drugs, reducing their cytotoxicity and degradation, providing prolonged action after administration, and improving the delivery of the active substance to the target tissue/organ [13,45,46]. The interest in liposomes as drug delivery systems is also related to the relative simplicity of their preparation and their ability to encapsulate a wide range of drugs regardless of their hydrophobic properties, charge, size, and other physicochemical properties [45,46]. Several methods are used for drug encapsulation into liposomes [37,46,47,48]. One method is based on lipid dissolution in an organic solvent, mixing with an aqueous medium, and subsequent removal of the solvent. In another method, micellar structures are obtained in the form of mixed lipid–detergent systems, from which the detergent is removed by dialysis below its critical micelle concentration. However, in both cases, there is a risk of biomolecule denaturation due to the use of organic solvents or detergents [48]. Another well-known method involves hydrating a lipid film. In this method, phospholipids are dissolved in a suitable organic solvent (such as chloroform or ethanol) and then removed under reduced pressure to obtain the thin lipid film, which is then hydrated in a solution containing the substances to be encapsulated. This method provides biomolecule encapsulation in its functional form, but the encapsulation efficiency is usually low [48]. However, to enhance the encapsulation efficiency, additional steps in the preparation of liposomes can be employed, such as freeze–thaw cycles. This approach is supported by previous studies [48,49], which demonstrated a significant improvement in encapsulation efficiency without substantial protein damage.

However, when developing methods for obtaining liposomes and their intended clinical application, it is necessary to take into account a number of physicochemical characteristics, such as particle size, distribution, surface charge, morphology, and structure [45]. It is essential to obtain homogeneous (monodisperse) nanoparticles of a specific size in order to create safe and effective nanocarriers. The dispersity of liposomes significantly affects the elimination rate, drug distribution, and concentration in biological fluids and tissues, as well as determines the mechanism of cell internalization. PDI, also known as the heterogeneity index, is a number calculated from a two-parameter fit to the correlation data (the cumulant analysis). This numerical value ranges from 0.0 (indicating a perfectly homogeneous sample) to 1.0 (corresponding to a highly polydisperse sample with varying particle sizes). As with particle size, the PDI value depends on the liposomal composition and the method of liposome production and is very sensitive to the presence of aggregates or contaminants (dust, foreign objects). The expected PDI value for monodisperse samples should not exceed 0.3. In addition, the zeta potential plays a significant role in the stability of liposomes, their circulation in the bloodstream, interactions with proteins and cells, the drug release profile, and biocompatibility. It serves as a crucial indicator of the liposomal surface charge and a measure of the electrostatic interactions (repulsion or attraction) between the particles [13,45,50,51,52,53]. An important step in obtaining and evaluating the liposomal drug formulation is the determination of the active substance’s encapsulation efficiency. This parameter allows the assessment of the successful rate of the active substance’s encapsulation within the liposomes and indicates the efficiency of this process [54]. The higher the encapsulation efficiency, the greater the amount of active ingredient that can be delivered to the target tissue/organ, which, in turn, can increase therapeutic efficacy. Another important aspect is the stability of liposomes. The physical and chemical instability of liposomes can lead to undesirable side effects and reduced therapeutic efficacy [9,55].

The evaluation of these characteristics on the properties of the liposomal drug formulation allows for determining optimal parameters for specific biomedical purposes and contributes to the improvement in their biocompatibility, reducing side effects and enhancing therapeutic efficacy.

Previously, a number of studies have been conducted in order to obtain a liposomal drug formulation of G-CSF [10,12,13]. For instance, J. Meyer et al. (1994) demonstrated the ability to encapsulate between 2 and 90% of G-CSF into liposomes using various methods [10]. However, the authors noted significant differences in the diameter of the obtained liposomes, which directly influenced the encapsulation efficiency. For instance, with an average liposome diameter of 253 nm (PDI = 0.3), the encapsulation efficiency was only 2%. When the diameter was increased to 343 nm (PDI = 0.6), the encapsulation efficiency reached 30%. Larger liposomes with diameters of 765 nm and 783 nm (PDI = 0.5 and 0.4, respectively) exhibited significantly higher encapsulation efficiencies of 80% and 90%. Similar results were obtained in a study conducted by F. Kiafar et al. (2016), as the encapsulation efficiency of G-CSF was observed to be around 50–60% for liposomes with sizes of 600–650 nm, obtained by thin-film hydration based on dipalmitoylphosphatidylcholine/cholesterol [12]. The overall encapsulation efficiency of G-CSF ranged from 39.3 to 73.7%. At the same time, liposomes sized less than 600 nm in diameter had encapsulation efficiency of G-CSF less than 50% [12]. These results confirm that the encapsulation efficiency of G-CSF in liposomes can vary and depends on several factors, including liposome composition, size, and conditions of their preparation [10,12]. It is worth noting that the above-mentioned studies did not provide detailed information on the physical stability of the obtained nanoparticles during prolonged storage.

In a study conducted by S. Lewicki et al. (2020), a protocol was developed for encapsulating two types of cytokines, granulocyte–macrophage colony-stimulating factor (GM-CSF) and G-CSF, into liposomes to preserve their activity over a long period of time [13]. However, the authors encapsulated only 12–13% of the total cytokine amount. They found that almost 50% of the cytokines (GM-CSF—47.4% and G-CSF—42.9%) were encapsulated in liposomes with sizes of 400–600 nm, while only 32% of GM-CSF and 28.3% of G-CSF were encapsulated in liposomes with sizes of 200–400 nm. Liposomes ≤200 nm contained less than 20% of the total amount of cytokines (G-CSF—17.6% and GM-CSF—15.3%) [13]. The authors attribute the low encapsulation efficiency to the absence of additional steps, such as freeze–thaw cycles and the intensive mixing or shaking of dispersions, which can significantly increase the degree of encapsulation [13].

According to the results of our study, G-CSF-loaded liposomes of regular round, oval, or droplet shapes were obtained with an average particle size of 161.9 ± 9.9 nm. They were homogeneous, had a low PDI value (less than 0.3), and were characterized by a high encapsulation efficiency of the active substance—52.37 ± 3.64% (Table 1). The obtained G-CSF-loaded liposomes were stable for at least 2 months of storage at a temperature of 4 °C and relative humidity of 60%. This indicates the successful encapsulation of G-CSF within the liposomes and demonstrates the effectiveness of additional steps (intensive shaking and five repeated cycles of freezing–thawing) during their production. In addition, the excess of PC (G-CSF ratio 1:100 by weight) could have influenced the relatively high-efficiency encapsulation of G-CSF within the liposomes.

The present study also utilized an MTT assay to assess the safety of G-CSF-loaded liposomes and revealed its dose-dependent effect on human dermal fibroblasts with no signs of cytotoxicity.

A comprehensive understanding of the obtained liposome efficacy in the biomedical field requires further studies, including in vivo. To assess the universality and biological activity of liposomes across different tissues and organ systems, it is also important to expand the functional analysis by including various cell types. This approach will contribute to the better characterization of their therapeutic potential and safety.

## 4. Materials and Methods

### 4.1. Materials

Lyophilized G-CSF was supplied by Bio Invest Scientific Technology Centre Llc (Moscow, Russia). The following reagents were used in this study: soy phosphatidylcholine Lipoid S-100 (PC, Lipoid, Ludwigshafen, Germany, CAS: 97281-47-5, composition of fatty acids: palmitic acid—15%; stearic acid—3%; oleic acid—12%; linoleic acid—62%; linolenic acid—5%) and phosphate-buffered saline (PBS, Rosmedbio Ltd., Saint Petersburg, Russia).

All solvents were of analytical grade and used as received from CHIMMED (Moscow, Russia).

### 4.2. Preparation of Granulocyte Colony-Stimulating Factor in Phosphate-Buffered Saline Solution and Granulocyte-Colony-Stimulating-Factor-Loaded Liposomes

#### 4.2.1. Granulocyte Colony-Stimulating Factor Solution

A solution of G-CSF (0.5 mg/mL) was prepared by dispersing the drug powder in PBS, aliquoted, and stored at −80 °C until use.

#### 4.2.2. Granulocyte-Colony-Stimulating-Factor-Loaded Liposomes

Liposomes were prepared with a thin-film hydration method [47]. Briefly, the solution of phosphatidylcholine (50 mg/mL) in chloroform was evaporated at a temperature of 30 ± 2 °C under a pressure of 40–45 mbar (Hei-Vap Advantage ML/G3B, Heidolph, Schwabach, Germany) to obtain a thin dry film. Following this, the thin dry film was dried under high-vacuum conditions (20–25 mbar) until a constant weight was achieved. For the preparation of the G-CSF-loaded liposomes’ suspension with a phospholipid concentration of 50 mg/mL, the lipid film was dispersed in PBS containing G-CSF (0.5 mg/mL, pH 7.4) by continuous mixing for 30 min until the thin lipid film was completely dispersed. To obtain uniformly shaped liposomes, the dispersion underwent five freezing–thawing cycles and was processed by sonication using an ultrasonication bath (3DT, Stegler, Moscow, Russia) for 10 min (40 kHz, 3 times). The freeze–thaw method we used involves rapid freezing by immersion of the flask in liquid nitrogen for 2 min, followed by slow thawing, which is conducted by immersing and gently shaking the tube in a water bath at 30 °C for 2 min. The latest stage of obtaining liposomes was membrane filtration through a sterilizing filter with a pore size of 0.22 µm (Express^®^ PLUS PES membrane GPWP02500, Merck Millipore, Burlington, MA, USA). Blank liposomes were prepared by the above-mentioned method but without G-CSF in their content.

All tests on liposomes were performed after overnight incubation (4 °C). The characteristics of the obtained liposomes were evaluated using physicochemical methods, i.e., determination of particle size distribution, zeta potential, and encapsulation efficiency; cryogenic transmission electron microscopy; and the in vitro effect of G-CSF-loaded liposomes on the metabolic activity of human dermal fibroblasts, followed by statistical data analysis.

### 4.3. Determination of Particle Size, Polydispersity Index, and Zeta Potential

The particle size distribution (mean hydrodynamic diameter and polydispersity index (PDI)) and zeta potential of the liposomes were measured using dynamic and electrophoretic light scattering on the Delsa Nano C (Beckman Coulter, Inc., Brea, CA, USA) at 25 °C. All measurements were performed in triplicate.

All samples were diluted in PBS (pH 7.4) at a ratio of 1:25. Then, 1 mL of the diluted solution was transferred to a polystyrene cuvette (a High-Concentration Zeta Potential Cell was used for measuring zeta potential). The measurement data were analyzed using the Delsa Nano UI Software Version 3.2 from Beckman Coulter Inc. (Brea, CA, USA).

### 4.4. Determination of the Encapsulation Efficiency of Granulocyte Colony-Stimulating Factor

The encapsulation efficiency of G-CSF was analyzed using high-performance liquid chromatography (HPLC) in reversed-phase mode (RP-HPLC) with an LC-20 Prominence HPLC system (Shimadzu, Kyoto, Japan). Chromatographic separation was performed on a Zorbax RX-C18 Analytical column (4.6 × 250 mm, 5 μm, Agilent, Santa Clara, CA, USA) according to the method described in [12]. A UV spectrophotometer was used as the detector. The mobile phases were composed as follows: Mobile phase A: acetonitrile (ACN) (49.9%) + water (50%) + trifluoroacetic acid (TFA) (0.1%). Mobile phase B: ACN (95%) + water (5%) + TFA (0.1%). Separation was performed in gradient mode at a flow rate of 1 mL/min, with detection at 215 nm and a set thermostat temperature of 65 °C. The corresponding conditions are listed in Table 4.

To avoid contamination from the G-CSF not encapsulated in the liposomes, the G-CSF-loaded liposome suspension was centrifuged at 10,000× *g* for 10 min (Gyrozen Mini Microcentrifuge, Gyrozen, Gimpo, Republic of Korea). The supernatant was collected, and the pellet was resuspended in PBS. This procedure was repeated three times.

The supernatant and pellet after centrifugation were analyzed for G-CSF content. The separated liposomes were lysed using 40% methanol. The linearity range was 1–50 µg/mL.

The percentage of encapsulation efficiency (EE%) of G-CSF was calculated using Equation (1):(1)$$EE\%=\frac{C_{p}}{C_{p}+C_{s}}\times 100\%,$$
where *C_p_* is the measured concentration of G-CSF in the pellet, and *C_s_* is the concentration of G-CSF in the supernatant.

### 4.5. Determination of the Morphology of Nanoparticle by Cryogenic Transmission Electron Microscopy

For cryogenic electron microscopy (Cryo-EM), 20 µL of the liposome suspension was applied to a 300-mesh copper grid with a perforated carbon coating (cat. no. 01895-F, Ted Pella, Inc., Redding, CA, USA), which had been pre-treated with plasma cleaning for 60 s to enhance hydrophilicity using an HDT-400 (JEOL Ltd., Akishima, Tokyo, Japan).

To remove excess solution, the grid was gently blotted from the reverse side with filter paper for 5 s, allowing a capillary film of solution to form within the perforations of the carbon film. Immediately after blotting, the grid was rapidly frozen by plunging into liquid ethane using a Leica EM GP cryopreparation station (Leica, Milan, Italy).

In cold nitrogen vapor, the grid was transferred from liquid ethane into liquid nitrogen and mounted into a Gatan 914 cryo-holder (Gatan, Inc., Pleasanton, CA, USA), cooled with liquid nitrogen at −181 °C, and inserted into the microscope. Visualization was performed at −120 °C with an accelerating voltage of 200 kV using a JEM-2100 HC transmission electron microscope (JEOL Ltd., Akishima, Tokyo, Japan) equipped with a NanoSprint43 Mk-II camera (Advanced Microscopy Techniques, Woburn, MA, USA).

### 4.6. Physical Stability Assessment of Granulocyte-Colony-Stimulating-Factor-Loaded Liposomes

The storage physical stability of G-CSF-loaded liposomes was evaluated 14, 30, and 60 days after their preparation. After each interval time, the samples were evaluated for the particle size, PDI, and zeta potential of liposomes. Formulations were stored as liquid suspensions at 4 °C and 60% humidity (*n* = 5) and set in place to determine the physical stability of the formulations over the storage period.

### 4.7. Assessment of Cell Viability

The assessment of the effect of the investigated drugs on the viability of cultured cells was performed using the MTT assay according to a standard protocol. The MTT assay is a colorimetric method that utilizes the tetrazolium salt MTT (3-(4,5-dimethylthiazol-2-yl)-2,5-diphenyltetrazolium bromide) and is based on the ability of mitochondrial dehydrogenases in living cells to reduce the colorless MTT reagent to purple crystalline formazan. The amount of formazan extracted from the cells with dimethyl sulfoxide (DMSO) solution is measured spectrophotometrically. This method enables the quantitative assessment of the impact of various factors on cells compared to the control.

In this study, human dermal fibroblasts were used as the cellular test system. These fibroblasts were isolated from human eyelid skin biopsies at the Institute of Cytology, Russian Academy of Sciences (St. Petersburg, Russia).

To evaluate the effect of the investigated drugs on cell viability, cells were seeded in 96-well plates at a density of 8 × 10^3^ cells per well in 100 µL of culture medium. After 24 h of cell adhesion, the standard culture medium was replaced with model media containing a solution of G-CSF, G-CSF-loaded liposomes, or empty liposomes at concentrations of 0.8, 1.6, and 3.2 µM, with corresponding adjustments based on the encapsulation efficiency. The cells were cultured at 37 °C in a CO_2_ incubator (Sanyo Co., Ltd., MCO-19AIC, Osaka, Japan) under a 5% CO_2_ atmosphere for 24 h. During cultivation, the morphological state of the cells was evaluated using an inverted microscope (ECLIPSE TS100, Nikon Instruments, Tokyo, Japan) equipped with a digital camera.

At the end of the cultivation period, the culture medium was replaced with 100 µL of medium containing MTT (Sigma-Aldrich, Saint Louis, MO, USA) (0.5 mg/mL in PBS) at 10% of the total volume. The plate was incubated in a CO_2_ incubator for 2 h, after which the MTT-containing medium was removed from the wells, and 100 µL of DMSO solution was added to each well.

The optical density of the resulting formazan solution in DMSO was measured using a Fluorofot “Charity” analyzer (Probanauchpribor, St. Petersburg, Russia) at a wavelength of 570 nm with a reference wavelength of 630 nm. The control consisted of cells cultured under standard conditions at 37 °C in a CO_2_ incubator with a 5% CO_2_ atmosphere in DMEM/F12 (Biolot, Saint Petersburg, Russia) supplemented with 10% fetal bovine serum (FBS, Gibco, Grand Island, NY, USA) and 1% penicillin–streptomycin (Gibco, Grand Island, NY, USA).

A reduction in cell viability of more than 30% was considered a cytotoxic effect.

### 4.8. Statistical Analysis

Statistical analysis was carried out using STATISTICA software version 7.1 (StatSoft, Tulsa, OK, USA), and statistical significance was set at level *p* ≤ 0.05. Statistical analysis was performed with a one-way ANOVA followed by Tukey’s multiple comparison test.

## 5. Conclusions

The present study resulted in a liposomal drug formulation of G-CSF, which exhibits high encapsulation efficiency and physical stability when stored at +4 °C and 60% humidity for at least 2 months. In vitro studies revealed a dose-dependent effect of G-CSF-loaded liposomes on human dermal fibroblasts and the lack of their cytotoxicity. To fully understand the effectiveness of the obtained liposomes in biomedical applications, further research is needed, including an evaluation of their effectiveness in vivo. Such studies will enable a deeper understanding of the behavior and efficacy of G-CSF-loaded liposomes under controlled conditions and in living systems. This will help us better understand their potential in biomedical applications, such as the treatment of various diseases, including those where immune system activation and granulocyte stimulation are important factors.

### Future Direction in Reproductive Medicine

Granulocyte colony-stimulating factor (G-CSF) is a cytokine that plays a crucial role in embryo implantation and normal pregnancy. It contributes to successful implantation by regulating the expression of genes involved in embryo adhesion, cell migration, tissue remodeling, and angiogenesis—key processes necessary for endometrial growth, embryo implantation, and subsequent placental development [26].

In recent years, intrauterine G-CSF treatment has gained significant attention despite its high cost. The majority of clinical trials have demonstrated promising outcomes, particularly in patients with thin endometrium and possibly those experiencing repeated implantation failure [3,26,28,56]. Additionally, stem cell therapy has shown potential in improving endometrial thickness in patients with Asherman’s syndrome or refractory thin endometrium, as well as in infertility treatment for women of reproductive age [57,58,59].

Long-term follow-up studies on children born from frozen embryos remain essential, particularly for women who have preserved their fertility by vitrifying oocytes due to oncological conditions such as breast, endometrial, or other gynecological malignancies [60,61]. These women often use their cryopreserved oocytes for parenthood after undergoing gonadotoxic treatments, including chemotherapy or radiotherapy [61,62,63].

Although the direct effect of G-CSF on the endometrium remains to be fully established, it is hypothesized that G-CSF facilitates endometrial repair through immune modulation, promoting the regeneration of damaged glands and stroma. Emerging evidence suggests that G-CSF enhances endometrial receptivity by stimulating trophoblast invasion via the upregulation of matrix metalloproteinase-2 (MMP-2) and vascular endothelial growth factor (VEGF), while concurrently reducing the expression of CD16/CD56 and increasing the expression of leukemia inhibitory factor (LIF). Furthermore, G-CSF may induce the mobilization of bone marrow-derived mesenchymal stem cells, which subsequently differentiate into epithelial cells, thereby supporting the regenerative processes of the endometrium

In this context, we hope and believe that G-CSF may serve as a valuable tool in addressing the challenges of declining birth rates, particularly in patients requiring fertility preservation due to medical conditions [3,56,64]. Liposomal delivery systems offer a potential solution by enabling controlled and sustained release of G-CSF, increasing its bioavailability, and minimizing systemic side effects. Encapsulation within liposomes protects G-CSF from rapid degradation, extends its circulation time, and ensures targeted delivery to the endometrial tissue. This controlled release mechanism may not only enhance therapeutic outcomes but also reduce the frequency of administration, thereby optimizing patient compliance and treatment efficacy.

## Figures and Tables

**Figure 1 ijms-26-02689-f001:**
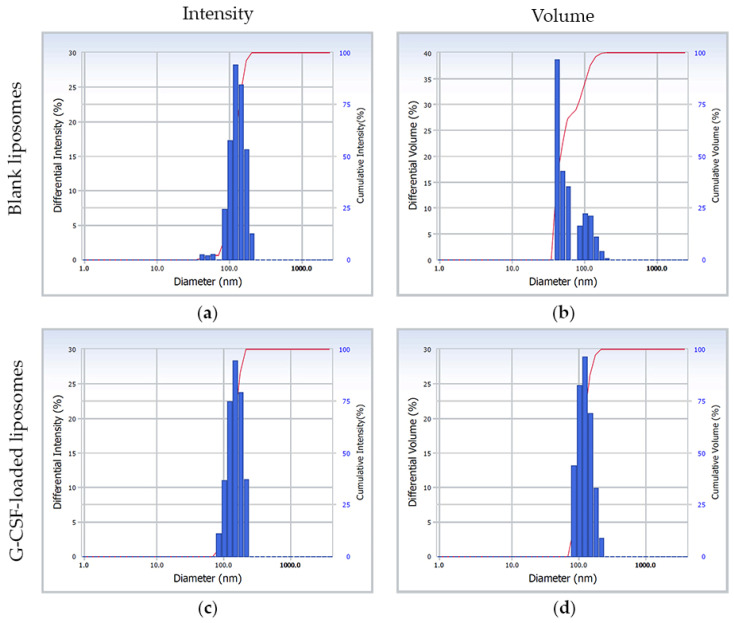
Intensity-weighted diameter distribution (**a**,**c**) and volume-weighted radius distribution (**b**,**d**) obtained from the light scattering analysis of the blank and granulocyte-colony-stimulating-factor-loaded liposomes. G-CSF—granulocyte colony-stimulating factor.

**Figure 2 ijms-26-02689-f002:**
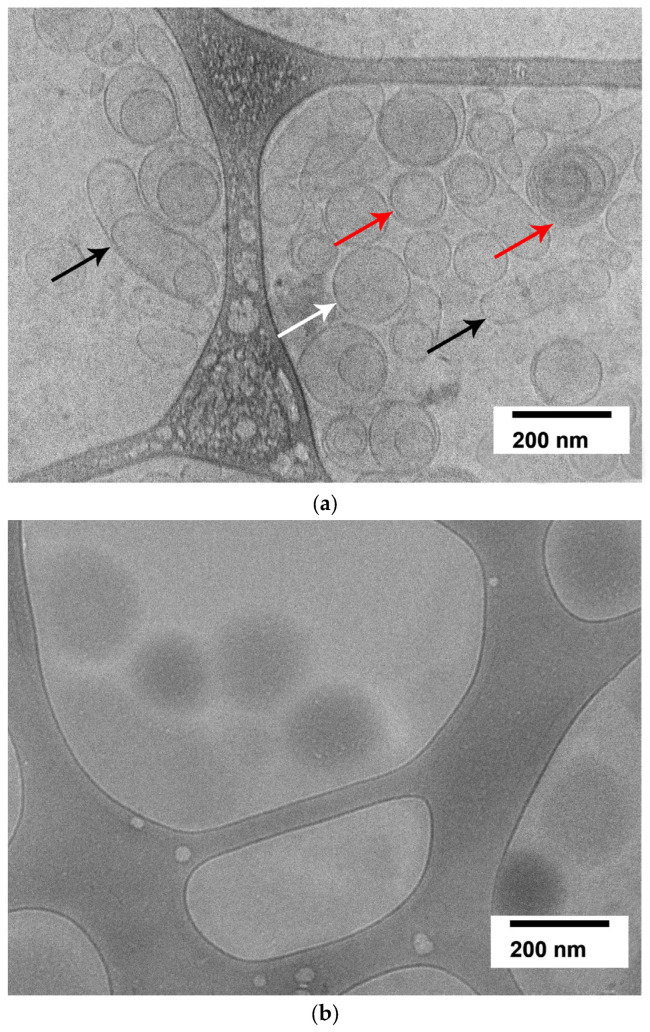
Cryogenic transmission electron microscopy of (**a**) blank liposomes and (**b**) G-CSF-loaded liposomes. The cryo-EM image of liposomes primarily shows spherical unilamellar vesicles in the size range of ~50–200 nm (white arrow), although some multilamellar vesicles (red arrow) and vesicles of droplet shapes (black arrow) were also present. Scale bar = 200 nm.

**Figure 3 ijms-26-02689-f003:**
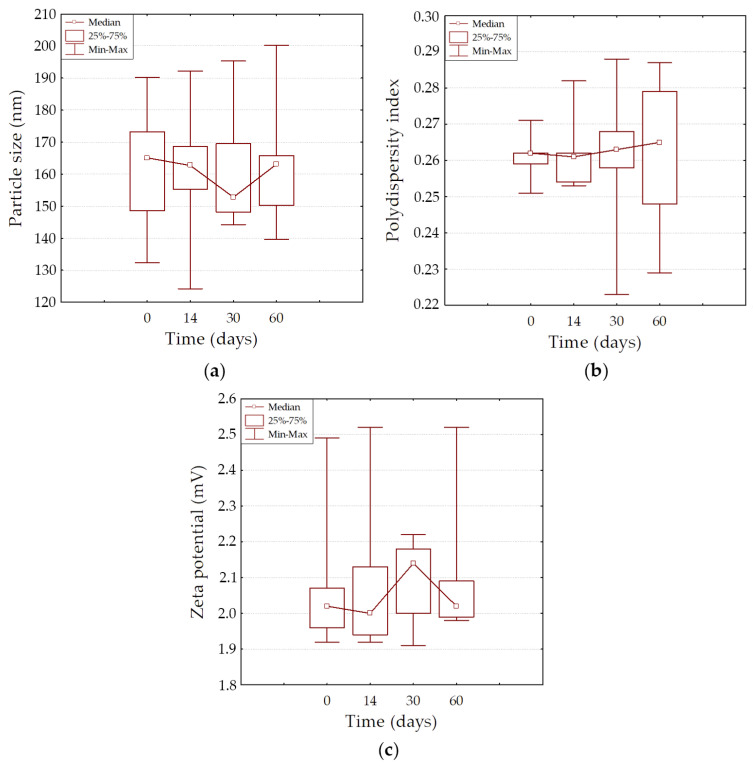
Physical stability of granulocyte-colony-stimulating-factor-loaded liposomes during long-term storage. Time influence during storage on particle size (**a**), polydispersity index (**b**), and zeta potential (**c**) of granulocyte-colony-stimulating-factor-loaded liposome.

**Figure 4 ijms-26-02689-f004:**
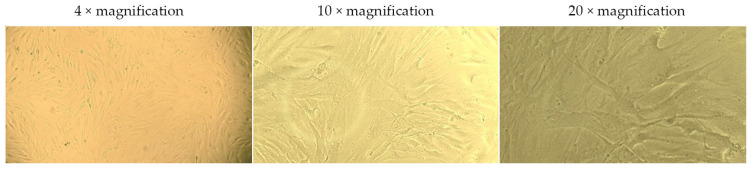
Morphology of dermal fibroblasts cultured for 24 h in a standard nutrient medium (control). The images correspond to the magnification levels indicated in the scale bar of the drawing: 4×, 10×, and 20×.

**Figure 5 ijms-26-02689-f005:**
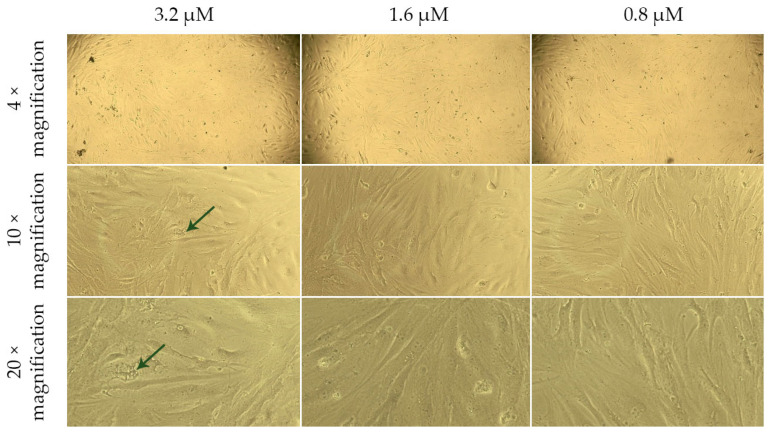
Morphology of dermal fibroblasts after 24 h of cultivation in model nutrient media containing G-CSF solution at concentrations of 3.2, 1.6, and 0.8 µM. The top panel displays 4× magnification, the middle panel shows 10× magnification, and the bottom panel presents 20× magnification.

**Figure 6 ijms-26-02689-f006:**
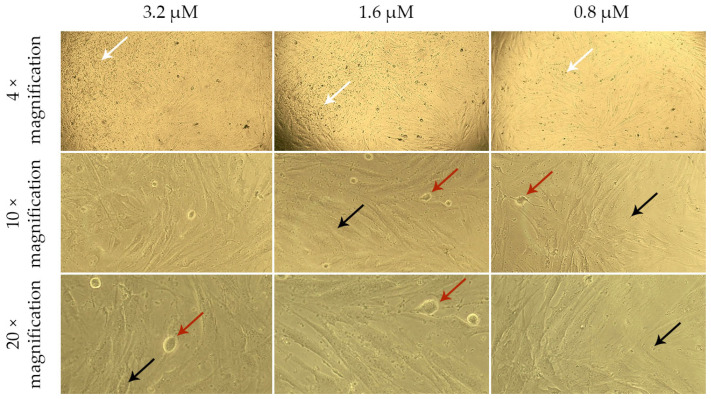
Morphology of dermal fibroblasts after 24 h of cultivation in model nutrient media containing liposomes with G-CSF at concentrations of 3.2, 1.6, and 0.8 µM. The top panel displays 4× magnification, the middle panel shows 10× magnification, and the bottom panel presents 20× magnification.

**Figure 7 ijms-26-02689-f007:**
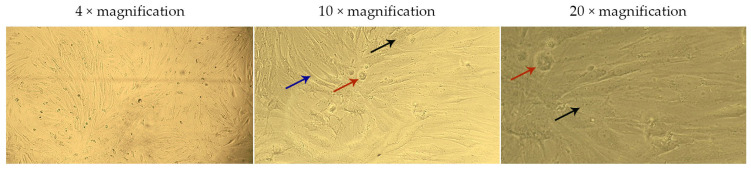
Morphology of dermal fibroblasts after 24 h of cultivation in model nutrient media containing blank liposomes. The images correspond to the magnification levels indicated in the scale bar of the drawing: 4×, 10×, and 20×.

**Figure 8 ijms-26-02689-f008:**
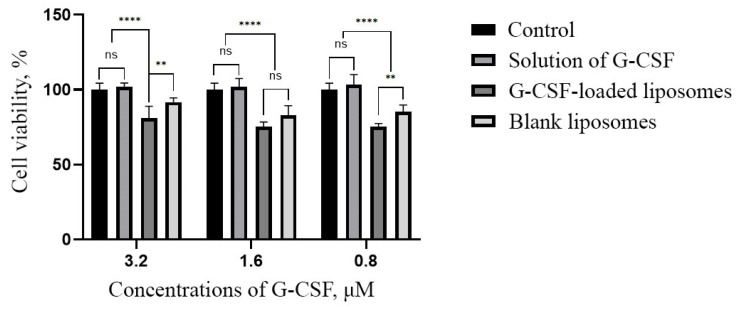
Viability of dermal fibroblasts after 24 h of cultivation in model nutrient media containing G-CSF solution, G-CSF-loaded liposomes, or blank liposomes at concentrations of 3.2, 1.6, and 0.8 µM. MTT assay. ns, not significant, **, *p*-value <0.01, ****, *p*-value < 0.0001.

**Table 1 ijms-26-02689-t001:** Characteristics of blank and granulocyte-colony-stimulating-factor-loaded liposomes.

Liposome’s Type	PC, mg/mL	G-CSF, mg/mL	PDI	Size, nm	ZP, mV	EE, %
Blank liposomes	50	–	0.271 ± 0.05	143.7 ± 7.7	–2.11 ± 0.11	–
G-CSF-loaded liposomes	50	0.5	0.261 ± 0.03	161.9 ± 9.9	+2.09 ± 0.10	52.37 ± 3.64

Key: PC—phosphatidylcholine; G-CSF—granulocyte colony-stimulating factor; PDI—polydispersity index; ZP—zeta potential; EE—encapsulation efficiency.

**Table 2 ijms-26-02689-t002:** Physicochemical characteristics of granulocyte-colony-stimulating-factor-loaded liposomes during long-term storage.

Liposome’s Type	Mean	Std. Err.	Std. Dev.	Median	Percentile 25%	Percentile 75%	Minimum	Maximum
Particle size, nm								
0 days	161.9	±9.9	±22.3	165.1	148.6	173.2	132.3	190.2
14 days	160.6	±11.0	±24.6	162.7	155.3	168.6	124.2	192.2
30 days	162.0	±9.4	±20.9	152.8	148.2	169.5	144.3	195.3
60 days	163.8	±10.2	±22.9	163.1	150.3	165.8	139.7	200.3
PDI								
0 days	0.261	±0.003	±0.007	0.262	0.259	0.262	0.251	0.271
14 days	0.262	±0.005	±0.011	0.261	0.254	0.262	0.253	0.282
30 days	0.260	±0.011	±0.024	0.263	0.258	0.268	0.223	0.288
60 days	0.262	±0.010	±0.023	0.265	0.248	0.279	0.229	0.287
Zeta potential, mV								
0 days	2.09	±0.10	±0.23	2.02	1.96	2.07	1.92	2.49
14 days	2.10	±0.11	±0.25	2.00	1.94	2.13	1.92	2.52
30 days	2.09	±0.06	±0.13	2.14	2.00	2.18	1.91	2.22
60 days	2.12	±0.10	±0.23	2.02	1.99	2.09	1.98	2.52

Key: Std. Err.—standard error; Std. Dev.—standard deviation; PDI—polydispersity index.

**Table 3 ijms-26-02689-t003:** Viability of dermal fibroblasts cultured for 24 h under standard conditions and in model nutrient media containing G-CSF solution, G-CSF-loaded liposomes, and blank liposomes.

Group	Cell Viability, %				
	3.2 μM	1.6 μM	0.8 μM	Control	Blank Liposomes
Solution of G-CSF	101.75 ± 2.67	101.87 ± 5.56	103.22 ± 6.79	100.00 ± 4.29	82.81 ± 6.53
G-CSF-loaded liposomes	80.99 ± 7.98	75.44 ± 2.95	75.15 ± 2.31		

**Table 4 ijms-26-02689-t004:** Gradient elution conditions in RP-HPLC for the analysis of encapsulation efficiency of G-CSF.

Time, min	Mobile Phase A, % *v*/*v*	Mobile Phase B, % *v*/*v*
0–4	92	8
4–19	92 to 72	8 to 28
19–19.1	72 to 0	28 to 100
19.1–21	0	100
21–21.1	0 to 92	100 to 8
21.1–30	92	8

## Data Availability

The data that support the findings of this study are available upon request from the corresponding author. The data are not publicly available due to privacy or ethical restrictions.

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
