# Peer review of "Formulation and In Vitro Characterization of Granulocyte-Colony-Stimulating-Factor-Loaded Liposomes: Future Prospective in Reproductive Medicine"

_ijms, 2025, doi:10.3390/ijms26062689_

Round 1
Reviewer 1 Report (New Reviewer)
Comments and Suggestions for Authors
In introduction, the authors evidenced the potential of G-CSF as a treatment for infertility but the study could be improved by adding some molecular mechanisms to explain G-CSF role in reproductive medicine.
Also, in introduction could be introduced a comparative analysis of liposomes and other types of systems (hydrogels, nanoparticles, etc.) for delivering of G-CSF. Why were chosen in drug delivery for reproductive applications?
What is the relevance of the viability studies on endometrial tissue if the cells used were from dermal fibroblasts? Also, the authors should detail how the liposomal formulation can interact with cellular components and what is the cause of the reduced viability.
Author Response
Please see the attachment

Reviewer 2 Report (New Reviewer)
Comments and Suggestions for Authors
This study is quite interesting and provides the fabrication and stability characterization of GCSF-loaded liposomes. Furthermore their compatibility was tested against dermal fibroblasts using state of the art MTT assays. Certainly, the evidence provided calls for more fundamental, biophysical studies to understand the mechanisms of the effects observed. This paper deserves to be published after the following points are addressed:
- In order to obtain small liposome with small polydispersity extrusion is typically carried out. The authors use sonication, would extrusion work for passive loading? When they speak about membrane filtration do they refer to extrusion? What is the pore size used?
- Table 2, too many significant digits of the reported values, please correct according to the uncertainties displayed
- Table 3, too many significant digits of the reported values, please correct according to the uncertainties displayed
- The Y-axis in Figure 5 panels (size, PDI and zeta potential) should be changed to a larger scale, to help the reader realize that liposomes are quite stable over time.
- Figure 4 needs to be better explained, justifying the difference in contrast between blank and loaded liposomes. What is the network-like structure seen? Authors should specify that liposomes are dried
- Arrows or some kind of indication should be added to Figures 8 and 9 in order to highlight the difference in action of loaded and blank liposomes in topography and detachment and more explanation of the differences included in the text
- Table 4, too many significant digits of the reported values, please correct according to the uncertainties displayed
- The authors conclude that further studies are necessary to understand the action mechanism of these loaded liposomes. They should provide more specific type of studies that may help, specially in understanding their mechanism of insertion (fusion)? and how they affect cell detachment. For example, they can refer to QCM-D, which would be very useful in the study of liposome adsorption on model systems, see, L. Bar et al., QCM-D Study of the Formation of Solid-Supported Artificial Lipid Membranes: State-of-the-Art, Recent Advances, and Perspectives, Physica Status Solidi A 220, 2200625 (2023), as well as how loaded liposome might modulate cell detachment, see, for instance D. Yongabi et al., Ionic strength controls long-term cell-surface interactions – A QCM-D study of S. cerevisiae adhesion, retention and detachment, Journal of Colloid and Interface Science 585, 583 (2021)
Author Response
Please see the attachment

This manuscript is a resubmission of an earlier submission. The following is a list of the peer review reports and author responses from that submission.
Round 1
Reviewer 1 Report
Comments and Suggestions for Authors
The idea is interesting, and it has scientific appeal; however, the study seems somewhat preliminary. Few experiments and data were presented to support any conclusions or future applications of the formulation. Even though there are no in vivo studies, more in vitro studies could have been included to better support the conclusions.
Other major comments:
- What are the adverse effects of G-CSF that could limit its use?
- In "determination of the granulocyte-colony stimulating factor extinction coefficients": Is an r² = 0.9709 acceptable to proceed with the studies, or could it compromise the results? Was phosphatidylcholine the only blank? Could the lipid interfere with the absorption?
- Why was a liposome composed only of soybean phosphatidylcholine chosen? Liposomes made solely from soybean phosphatidylcholine (soy PC) have certain stability limitations, particularly when it comes to fusion, aggregation, and oxidation.
- What do you think caused the shift in the zeta potential when G-CSF was added to the blank liposome?
- For the encapsulation efficiency, why was centrifugation or ultracentrifugation not employed to separate free G-CSF from encapsulated G-CSF? Gel filtration adds unnecessary complexity to sample preparation.
- Figure 4 – Why didn't you use cryo-microscopy? TEM alone destroys the samples, and the figure looks poor; it is not possible to conclude that the vesicles are regular and spherical.
- Why was encapsulation efficiency (%EE) not evaluated in the stability study? Without assessing drug loss, it is premature to claim that the liposome remains stable for 2 months.
- Figure 6 isn’t very clear. It’s hard to see the differences mentioned in the text. Also, highlight statistical difference, and check if the differences are really significant—those high standard deviations make it questionable.
- Lines 245-250: Review the paragraph; there are some conceptual errors. In the lipid film hydration method, the bilayer is formed at the time of hydration. The %EE is low depending on the molecule being studied.
- Regarding the liposome preparation method: Was sonication alone insufficient to achieve the desired size? Could the freeze-thaw cycles have negatively impacted G-CSF encapsulation?
- Line 358: The phrase "All tests on liposomes were performed after overnight incubation (4°C)" needs clarification. Incubation with what?
Comments on the Quality of English LanguageEnglish should be revised and improved, especially in the introduction section.
Author Response
The idea is interesting, and it has scientific appeal; however, the study seems somewhat preliminary. Few experiments and data were presented to support any conclusions or future applications of the formulation. Even though there are no in vivo studies, more in vitro studies could have been included to better support the conclusions.
Other major comments:
- What are the adverse effects of G-CSF that could limit its use?
The side effects associated with G-CSF are demonstrated in detail in a recent review (R.S. Bumbăcea et al., 2024). The main adverse effects include: 1) Muscle pain was observed in 20-25% of patients; 2) Splenomegaly — 25% of patients recorded asymptomatic spleen enlargement, accompanied by abdominal or shoulder pain; 3) Cardiovascular complications, including arrhythmias, coronary and arterial thromboses, which could be related to G-CSF-induced hypercoagulation state. Cases of aortitis and large vessel vasculitis have also been reported; 4) Dermal complications, including necrotizing pyoderma, Sweet's syndrome, other neutrophilic dermatoses, granulomatous dermatitis, widespread folliculitis, cutaneous vasculitis, as well as acne and psoriasis exacerbations. These side effects may limit the use of G-CSF in certain clinical situations, especially in patients predisposed to vascular or dermatological issues.
Bumbăcea, R.S.; Udrea, M.R.; Ali, S.; Bojincă, V.C. Balancing Benefits and Risks: A Literature Review on Hypersensitivity Reactions to Human G-CSF (Granulocyte Colony-Stimulating Factor). Int. J. Mol. Sci. 2024, 25, 4807. https://doi.org/10.3390/ijms25094807
- In "determination of the granulocyte-colony stimulating factor extinction coefficients": Is an r² = 0.9709 acceptable to proceed with the studies, or could it compromise the results? Was phosphatidylcholine the only blank? Could the lipid interfere with the absorption?
The extinction coefficient r² = 0.9709 is sufficiently high and indicates a good reliability between experimental data and the model used to determine the effectiveness of G-CSF encapsulation. The obtained value is acceptable for further studies and should not significantly affect conclusion, although striving for a higher value (r² closer to 1) is always desirable. Regarding the phosphatidylcholine as a blank, we conducted studies to evaluate the influence of various lysing agents and the liposomes themselves on the spectral characteristics of the UV spectrum. As a result, the influence turned out being insignificant, thus, it can be considered during the experiment.
- Why was a liposome composed only of soybean phosphatidylcholine chosen? Liposomes made solely from soybean phosphatidylcholine (soy PC) have certain stability limitations, particularly when it comes to fusion, aggregation, and oxidation.
Liposomes composed solely of soy phosphatidylcholine (PC) were chosen due to their biocompatibility and well-studied properties. Soy PC (fatty acid composition: palmitic acid 15%, stearic acid 3%, oleic acid 12%, linoleic acid 62%, linolenic acid 5%) is a natural phospholipid that is readily available and is often used in the development of liposome-based systems. It forms stable membranes and actively interacts with biomolecules, including proteins such as G-CSF.
However, as you noted, liposomes made solely from soy PC have certain limitations. Such liposomes may be prone to fusion, aggregation and oxidation, resulting in reduced long-term stability. The absence of cholesterol or other stabilizing compounds makes the membranes more susceptible to structural changes during storage, especially when exposed to external factors such as temperature or reactive oxygen species. However, due to the chemical nature of the cholesterol, its addition leads to reduced active substance’s inclusion rate and increases the "rigidity" of the lipid bilayer (V.A. Shchelkonogov et al. (2021)).
Our research was investigating the stability of liposomal formulations made solely from soy phosphatidylcholine. We aimed to determine how effective such liposomes maintain their properties over time, despite the potential risks of fusion, aggregation and oxidation. The results will enable us to assess whether future modifications in liposome composition are necessary to enhance their stability.
Shchelkonogov, V.A., Darnotuk, E.S., Chekanov, A.V. et al. Liposomal Drug with Carnosine and Lipoic Acid: Preparation, Antioxidant and Antiplatelet Properties. BIOPHYSICS 66, 278–285 (2021). https://doi.org/10.1134/S0006350921020214
- What do you think caused the shift in the zeta potential when G-CSF was added to the blank liposome?
The minor shift in zeta potential observed in blank liposome upon G-CSF encapsulation is likely caused by the electrostatic interactions between the liposome’s lipid bilayer and the G-CSF molecule. Although G-CSF contains positively charged amino acids (less than 10% of its composition), under physiological pH (around 7.4, PBS) the majority of amino acid residues exist in zwitterion form, where the positive and negative charges are balanced. This should minimize the impact on the overall protein charge and its interaction with the liposome.
The minor recorded changes in zeta potential can be disregarded, as it was indicated (D.Y. Wang et al. (2020); M.C. Smith et al. (2017); Y. Nie et al. (2012); M. Campos-Martorell et al. (2016); M. Narenji et al. (2017)) that liposomes with zeta potential ranging from −10 to +10 mV are considered neutral. Therefore, it can be asserted that the obtained G-CSF-loaded liposomes remain neutral, similar to blank liposomes.
Wang, D.Y., van der Mei, H. C., Ren, Y., Busscher, H. J., & Shi, L. (2020). Lipid-Based Antimicrobial Delivery-Systems for the Treatment of Bacterial Infections. Frontiers in chemistry, 7, 872. https://doi.org/10.3389/fchem.2019.00872
Smith, M. C., Crist, R. M., Clogston, J. D., & McNeil, S. E. (2017). Zeta potential: a case study of cationic, anionic, and neutral liposomes. Analytical and Bioanalytical Chemistry, 409(24), 5779–5787. doi:10.1007/s00216-017-0527-z
Nie, Y., Ji, L., Ding, H., Xie, L., Li, L., He, B., Wu, Y., & Gu, Z. (2012). Cholesterol derivatives based charged liposomes for doxorubicin delivery: preparation, in vitro and in vivo characterization. Theranostics, 2(11), 1092–1103. https://doi.org/10.7150/thno.4949
Campos-Martorell, M., Cano-Sarabia, M., Simats, A., Hernández-Guillamon, M., Rosell, A., Maspoch, D., & Montaner, J. (2016). Charge effect of a liposomal delivery system encapsulating simvastatin to treat experimental ischemic stroke in rats. International journal of nanomedicine, 11, 3035–3048. https://doi.org/10.2147/IJN.S107292
Narenji, M., Talaee, M. R., & Moghimi, H. R. (2017). Effect of Charge on Separation of Liposomes upon Stagnation. Iranian journal of pharmaceutical research: IJPR, 16(2), 423–431.
- For the encapsulation efficiency, why was centrifugation or ultracentrifugation not employed to separate free G-CSF from encapsulated G-CSF? Gel filtration adds unnecessary complexity to sample preparation.
Centrifugation and/or ultracentrifugation for free G-CSF separation from encapsulated G-CSF may seem simpler than gel filtration. However, centrifugation, particularly ultracentrifugation, generates high forces leading to liposomes’ disruption or aggregation. In contrast, gel filtration is a gentler separation technique that minimizes mechanical stress on lipid structures. Gel filtration results in coherent separation of free G-CSF from encapsulated G-CSF based on differences in molecular sizes and complex interactions. Centrifugation may not provide the same degree of selectivity.
Thus, despite the additional complexity, gel filtration provides more reliable and safe separation of free and encapsulated G-CSF while preserving the chemical properties of both the liposomes and the protein. This is critically important for assessing the degree of the active compound’s incorporation.
In addition, the studies were conducted in different laboratories and, unfortunately, resources to gel filtration were limited in the laboratory where the biological properties were assessed.
- Figure 4 – Why didn't you use cryo-microscopy? TEM alone destroys the samples, and the figure looks poor; it is not possible to conclude that the vesicles are regular and spherical.
Cryo-microscopy is indeed the preferred method for the investigation of liposomes and other nanostructures, as it provides observation in their native state, avoiding risks associated with traditional transmission electron microscopy (TEM). Unfortunately, there was a lack of access to the specialized equipment necessary for cryo-microscopy in the laboratory where the study was conducted.
We recognize that cryo-microscopy would provide accurate data on the morphology of liposomes. Your comment is entirely justified, and we will strive to utilize this method in future studies in order to enhance the quality of images and accuracy of the results. However, at present, the implementation of this method is impossible due to the lack of appropriate equipment and infrastructure.
- Why was encapsulation efficiency (%EE) not evaluated in the stability study? Without assessing drug loss, it is premature to claim that the liposome remains stable for 2 months.
You are absolutely correct. The evaluation of encapsulation efficiency (%EE) plays a key role in confirming the stability of liposomal formulations. In addition to the aforementioned aspects, it should be noted that the degree of G-CSF encapsulation was also measured at 3 months of storage (data not shown in the main manuscript), and no significant changes in encapsulation efficiency were observed during the analysis. This indicates that the liposomes maintained their integrity and ability to retain the encapsulated substance over this period.
However, despite the positive results at 3 months period, it is important to conduct regular %EE assessments at all stages of storage for a more comprehensive confirmation of the long-term stability and retention of the encapsulated substance. Thus, in future studies, we plan to pay special attention to the dynamics of encapsulation efficiency at different time points to completely rule out the possibility of active substance loss.
- Figure 6 isn’t very clear. It’s hard to see the differences mentioned in the text. Also, highlight statistical difference, and check if the differences are really significant—those high standard deviations make it questionable.
We confirm that a reliable difference is indeed observed at basal respiration (4.30±0.07 for control vs. 5.84±0.12 for 10 mg/ml (mean ± SEM over 5 time points, p = 0.000011)) and after oligomycin administration (2.47±0.09 for control vs. 4.32±0.1 for 10 mg/ml (mean ± SEM over 5 time points, p = 0.000003)), which inhibits ATP synthesis.
The absence of a difference in subsequent stages, such as maximal respiration and inhibition of the respiratory chain, indicates a lack of increase in the number of mitochondria and significant proton leakage. This confirms the reversible nature of the effect in the context of cellular respiration.
We acknowledge that visual differences might be less noticeable on the graph; however, statistical results confirm the significance of the observed effects at the specified stages.
- Lines 245-250: Review the paragraph; there are some conceptual errors. In the lipid film hydration method, the bilayer is formed at the time of hydration. The %EE is low depending on the molecule being studied.
Corrected.
- Regarding the liposome preparation method: Was sonication alone insufficient to achieve the desired size? Could the freeze-thaw cycles have negatively impacted G-CSF encapsulation?
Sonication alone may be insufficient to achieve the desired liposome size, as this depends on the required particle diameter and uniform shape. While sonication effectively reduces liposome size, additional methods may be necessary to achieve a more uniform distribution (for the formation of homogeneous, uniform nanoparticles) and optimal size.
Freezing-thawing cycles are applied to improve encapsulation and the uniform distribution of proteins and other biological molecules within liposomes and nanoparticles. For instance, J.P. Colletier et al. (2002) observed an increase in the acetylcholinesterase encapsulation efficiency into liposomes depending on the number of freezing-thawing cycles without significant protein denaturation. Similar conclusions were drawn by A.C. Khayrani et al. (2024), who established that freezing and thawing increased the encapsulation efficiency of bovine serum albumin, as well as the particle size and PDI, to 81.63±0.00%, 180.70±0.87 nm, and 0.369±0.02, respectively, compared to transferosomes without freezing-thawing cycles (73.35±0.03%, 144.93±0.21 nm, and 0.202±0.02).
These results indicate that freezing-thawing cycles can enhance the degree of protein encapsulation, making them an important step in preparation of liposomes.
Colletier, J. P., Chaize, B., Winterhalter, M., & Fournier, D. (2002). Protein encapsulation in liposomes: efficiency depends on interactions between protein and phospholipid bilayer. BMC biotechnology, 2, 9. https://doi.org/10.1186/1472-6750-2-9
Khayrani, A.C., Fahmi, M., Nurhayati, R.W., Manas, N.H.A., Suhaeri, M., 2024. Effect of Freeze-Thaw Cycles Method to Transfersome Characteristics for Growth Protein Encapsulation. International Journal of Technology. Volume 15(2), pp. 267-278
- Line 358: The phrase "All tests on liposomes were performed after overnight incubation (4°C)" needs clarification. Incubation with what?
Thank you for your observation! We rephrased it for accurate description of the storage conditions. Here is a rephrased version: "All tests on liposomes were conducted after their storage at 4°C overnight."
Reviewer 2 Report
Comments and Suggestions for Authors
I would like to thank the authors for their work. I hope the authors will not be discouraged by this feedback, and will instead use it as a guide to improve their research.

The manuscript appears to have been written separately by at least three people with distinct styles. One is particularly proficient linguistically, while the other two, who appear to have taken care of the technical sections, appear to have difficulties with English. Nonetheless, there are factual and expression errors that need correcting. Table and Figure references and captions have mistakes which hinder understanding.
Author Response
Review of “Formulation and in vitro characterization of granulocyte colony-stimulating factor loaded liposomes”, authored by Kseniia V. Obedkova, Galina Kh. Safarian, Dmitry V. Tovpeko, Julian R. Ryzhov, Olesya N. Bespalova, and Natalya I. Tapilskaya.
Summary of work
The authors highlight a very important area of pharmaceutical and biomedical research, which is the development of drug formulations that not only enhance treatment efficacy, but also reduce side-effects. In this work, they attempt to make a contribution to granulocyte colony-stimulating factor (G-CSF)-based therapy. The authors describe the encapsulation of G-CSF in soy PC liposomes, ostensibly for the purpose of rendering the G-CSF less toxic and to prolong it’s activity. They also describe physicochemical and cytotoxicity characterisation of these liposomes.
- Aims
Although the authors provide a clear and succinct explanation of this field of research, the specific aims of their own work are not explicitly stated. It was necessary to infer their goals from the following statements in the abstract as well as introduction:
Lines 15-16: Encapsulation of G-CSF within liposomes may enhance its stability, bioavailability and controlled release properties.
This statement implies that they are trying to use encapsulation in liposomes to stabilise G-CSF, increase its systemic residence time, and to be able to decide where, when, and how fast it is released from the liposomes, in patients.
Lines: 81-83: However, commercially available forms of G-CSF have a short pharmacological effect, while their systemic administration can cause various side effects, limiting its effectiveness.
Here, the motivation for their work appears to be the fact that current formulations of G-CSF do not have a lasting treatment effect (implying fast clearance), cause various side effects when administered systemically, and are not effective in treatment (implying loss of stability).
Lines 84-86: Thus, the present study aimed to obtain a liposomal drug formulation of G-CSF and to evaluate its chemical and biological characteristics, including physicochemical stability and safety.
However, the stated aims immediately after the motives are to produce G-CSF-loaded liposomes and to ascertain their characteristics as stated above. The introduction then ends abruptly, without an explicit explanation of how their work would address the problems stated above, nor an elaboration of what is meant by “chemical and biological characteristics”, “physicochemical stability”, and “safety”.
These vague descriptions do not introduce, nor explain their methodology, which had to be precariously inferred from the Results, and later, in the Methods section.
Thank you for your comment. By "chemical and biological characteristics," "physicochemical stability," and "safety," we mean the investigation of key characteristics of liposomes, such as their size, polydispersity, encapsulation efficiency and storage stability, as well as the evaluation of biocompatibility and safety of liposomes in relation to fibroblasts.
We acknowledge that the objectives in the "Introduction" section could have been articulated more clearly; however, the primary aim of the "Introduction" is to provide a thorough description in the field of current research and briefly mention the main goal of the study without delving into the specifics of the applied methods. In contrast, the methods are described in greater detail in the "Materials and Methods" section.
- Novelty
B1. More importantly, the findings presented do not appear to add much to the existing literature.
As the authors themselves point out, the encapsulation of G-CSF in liposomes is not new (Lines 280-302).
Despite prior studies dedicated to the G-CSF encapsulation in liposomes, G-CSF-loaded liposomal formulations have not been implemented in clinical practice. This underscores the importance of further research aimed at improving the stability, encapsulation efficiency and safety of these formulations. Our study contributes to understanding how to optimize liposomal systems for G-CSF delivery, thereby bringing the possibility of its clinical application closer.
While the authors repeatedly emphasised the encapsulation efficiency of their protocol, describing it as high, it is by no means remarkable when compared to those reported by the literature that they themselves cite. In fact, the encapsulation efficiency of G-CSF that they report is a matter of concern, as will be elaborated on later.
We agree with your observation that the encapsulation efficiency in our study is not outstanding in comparison to those reported in the literature. However, it is important to note that other studies (J. Meyer et al. (1994); F. Kiafar et al. (2016); S. Lewicki et al. (2020)) focused on such parameters as polydispersity (PDI) and particle size in addition to encapsulation efficiency. For instance, J. Meyer et al. (1994), demonstrated the ability to encapsulate from 2% to 90% of G-CSF. However, significant differences in diameter of liposomes and their heterogeneity were also noted: "The mean diameter of liposomes prepared using method 1 was 253 nm (polydispersity=0.2). The mean diameter of liposomes prepared using method 2 was 343 nm (polydispersity=0.6). The mean diameters and polydispersities of lyposomes prepared using method 3 and 4 were 765 nm (0.5) and 783 nm (0.4), respectively." Similar results were reported by F. Kiafar et al. (2016) and S. Lewicki et al. (2020), as the particles’ size and heterogeneity increased with higher degrees of encapsulation.
In our study, the application of additional steps, including freeze-thaw cycles, resulted in the achievement of homogeneous liposomes up to 200 nm and a PDI < 0.3. This indicates that our liposomes, although having a more moderate encapsulation efficiency, are smaller in size and are highly homogenous, which are important factors for the stability and bioavailability of liposomal formulations.
B2. Furthermore, it is curious that the authors point out in lines 249-250 that gentle hydration usually results in low encapsulation efficiencies. And yet, they used precisely this technique to produce their G-CSF-loaded liposomes. Lines 299-302 suggests why, by explaining how repeated freeze-thawing could improve encapsulation efficiencies. However, this post-treatment is not novel, either.
Which leads one to conclude that it is the not the use of liposomes, nor their method of encapsulating G-CSF, that is novel in this work.
We agree that gentle hydration generally leads to low encapsulation efficiency, especially under standard conditions. However, we chose this method as it preserves the biological activity of G-CSF while minimizing the risk of protein denaturation, which is critically important for maintaining its therapeutic properties. To enhance encapsulation efficiency, we added freeze-thaw cycles. This approach was supported by previous studies (e.g., J.P. Colletier et al. (2002) and A.C. Khayrani et al. (2024)), which demonstrated significant improvement of encapsulation efficiency without substantial protein damage.
Thus, the application of gentle hydration was dictated by the need to preserve the functional activity of G-CSF, and the introduction of additional steps allowed increasing encapsulation efficiency to acceptable levels. The novelty of our study lies in the production of homogeneous G-CSF-loaded liposomes sized up to 200 nm and with PDI <0.3, as well as a detailed assessment of their stability over time, which is an important step toward the development of effective liposomal formulations. Our approach demonstrates that high homogeneity of liposomes and stable parameters can be achieved when encapsulating biologically active molecules such as G-CSF, bringing these systems closer to clinical application.
Nevertheless, we acknowledge that the methods we used to prepare liposomes are based on well-known and established techniques developed over decades, such as the lipid film hydration method and freeze-thaw cycles. Despite their widespread application, the combination of these methods under optimized conditions allowed us to achieve liposomal characteristics not previously reported in other studies.
Colletier, J. P., Chaize, B., Winterhalter, M., & Fournier, D. (2002). Protein encapsulation in liposomes: efficiency depends on interactions between protein and phospholipid bilayer. BMC biotechnology, 2, 9. https://doi.org/10.1186/1472-6750-2-9
Khayrani, A.C., Fahmi, M., Nurhayati, R.W., Manas, N.H.A., Suhaeri, M., 2024. Effect of Freeze-Thaw Cycles Method to Transfersome Characteristics for Growth Protein Encapsulation. International Journal of Technology. Volume 15(2), pp. 267-278
B3. Lines 289-291 appear to suggest that the works cited do not provide (detailed) information on long-term storage stability of G-CSF-loaded liposomes. Nor do their authors describe the biocompatibility of the G-CSF-loaded liposomes. It is difficult to imagine, that none of this previous research had not, at least, evaluated the effect of adding G CSF-loaded liposomes to cells.
Without consulting these cited reports (although I have done so), a generous assumption would be that this manuscript’s main contribution is the detailed investigation of the long-term storage stability (which is assumed to be novel) of this formulation (which is not novel), as well as the finding of their relatively benign effect on human dermal fibroblasts (which is assumed to be novel).
Even if we grant these novelties, the conclusions that the authors present are difficult to accept as useful and sometimes hard to accept as valid. This will be explained below.
A study conducted by J. Meyer et al. (1994) primarily focused on the technology of G-CSF encapsulation in liposomes and evaluated the active substance’s release both in vitro and in vivo. However, authors did not provide detailed information regarding the long-term stability of liposomes during storage and their biocompatibility.
The study by F. Kiafar et al. (2016) proposed a prediction model on characteristics of filgrastim (G-CSF)-loaded liposomes. This model was used to create optimal liposome formulations with improved properties, such as size and encapsulation efficiency. However, it is important to note that the evaluation of liposomes’ biological properties, as well as their long-term stability was not conducted.
Only S. Lewicki et al. (2020) performed a comprehensive analysis of G-CSF-loaded liposomes, including the assessment of their biological activity and stability during storage over a period of 28 days under various conditions. However, the described liposomes significantly differ from ours in particle size.
- Methods and Conclusions
In order to make robust conclusions, the methods of sample prepara6on as well as those for obtaining the suppor6ng evidence have to be adequate. Regrettably, the technical sections are fraught with inaccuracies. Here are samples of the many problems that compromise the strength of the authors’ arguments.
Preparation and physicochemical characterisation of G-CSF-loaded liposomes
C1. Line 119 describes liposomes as being produced by “passive loading”. This is a mistake. Passive loading is what was used to encapsulate the G-CSF, not to produce liposomes.
Corrected in the main text.
C2. Lines 248-249 describe lipid bilayers as being “hydrated in a solution containing the encapsulated substances”. This is a mistake since there were no encapsulated substances in the hydra6on buffer. Perhaps the authors mean “hydrated in a solu6on containing the substances to be encapsulated”?
Corrected in the main text.
C3. Line 343 describes the PC thin film as liposomes. This is a clear mistake and needs to be amended.
Corrected in the main text.
C4. Lines 344-346 describe hydration of the thin film by G-CSF in PBS, but also with “a buffer solution” in line 347. This double-men6on of the hydra6on buffer is confusing and suggests that there are two dis6nct types. These statements need to be rephrased.
Corrected in the main text.
C5. Lines 354-356 explain that unilamellar vesicles were obtained by filtration through a syringe filter. This is usually not enough to create unilamellarity. Typically, multiple extrusions under high pressure using an extruder is needed for this, as well as to reduce the polydispersity of a liposome preparation. If their liposomes were really unilamellar, it was probably due to the repeated freeze-thawing. However, the authors do not present evidence of unilamellarity at all.
We concur with your observation that filtration through a syringe filter alone is not always sufficient to obtain unilamellar liposomes. However, our study additionally utilized sonication, which, according to the literature, promotes unilamellarity (R. Silva et al. (2010); J.-S. Yun et al. (2023); C.F. de Freitas et al. (2019)). Nevertheless, further research is required to fully confirm this assertion, such as the application of cryo-microscopy and the selection of appropriate conditions for sonication.
Silva, R., Ferreira, H., Little, C., & Cavaco-Paulo, A. (2010). Effect of ultrasound parameters for unilamellar liposome preparation. Ultrasonics sonochemistry, 17(3), 628–632. https://doi.org/10.1016/j.ultsonch.2009.10.010
Yun, J.-S.; Hwangbo, S.-A.; Jeong, Y.-G. Preparation of Uniform Nano Liposomes Using Focused Ultrasonic Technology. Nanomaterials 2023, 13, 2618. https://doi.org/10.3390/nano13192618
de Freitas, C. F., Calori, I. R., Tessaro, A. L., Caetano, W., & Hioka, N. (2019). Rapid formation of Small Unilamellar Vesicles (SUV) through low-frequency sonication: An innovative approach. Colloids and surfaces. B, Biointerfaces, 181, 837–844. https://doi.org/10.1016/j.colsurfb.2019.06.027
C6. In Table 2, the concentration of PC used should be indicated for both sample types in order to reduce confusion.
Corrected in the main text.
C7. Lines 131-134 state that all the liposome formula6ons they’d produced were neutrally charged, and also explained that neutrally charged liposomes are desirable for use in systemic administration. However, although PC is a neutral molecule as a whole, their unloaded liposomes displayed a nega6ve surface charge while their G-CSF loaded ones had a posi6ve surface charge. It is explicitly pointed out that interac6on with nega6vely-charged serum proteins, which would likely occur for the surface-positive G-CSF liposomes, is undesirable. All this appears to contradict lines 131-134. There is also no explana6on for the difference in surface charge observed between the liposome types.
We acknowledge that phrasing in lines 131-134 may raise questions, and we appreciate your attention to this aspect. However, the minor changes in zeta potential that were recorded can be disregarded, as studies (D.Y. Wang et al. (2020); M.C. Smith et al. (2017); Y. Nie et al. (2012); M. Campos-Martorell et al. (2016); M. Narenji et al. (2017)) indicate that liposomes with zeta potentials ranging from -10 to +10 mV are considered electrically neutral. Thus, it can be asserted that the obtained G-CSF-loaded liposomes remain neutral, similar to the blank ones.
Wang, D.Y., van der Mei, H. C., Ren, Y., Busscher, H. J., & Shi, L. (2020). Lipid-Based Antimicrobial Delivery-Systems for the Treatment of Bacterial Infections. Frontiers in chemistry, 7, 872. https://doi.org/10.3389/fchem.2019.00872
Smith, M. C., Crist, R. M., Clogston, J. D., & McNeil, S. E. (2017). Zeta potential: a case study of cationic, anionic, and neutral liposomes. Analytical and Bioanalytical Chemistry, 409(24), 5779–5787. doi:10.1007/s00216-017-0527-z
Nie, Y., Ji, L., Ding, H., Xie, L., Li, L., He, B., Wu, Y., & Gu, Z. (2012). Cholesterol derivatives based charged liposomes for doxorubicin delivery: preparation, in vitro and in vivo characterization. Theranostics, 2(11), 1092–1103. https://doi.org/10.7150/thno.4949
Campos-Martorell, M., Cano-Sarabia, M., Simats, A., Hernández-Guillamon, M., Rosell, A., Maspoch, D., & Montaner, J. (2016). Charge effect of a liposomal delivery system encapsulating simvastatin to treat experimental ischemic stroke in rats. International journal of nanomedicine, 11, 3035–3048. https://doi.org/10.2147/IJN.S107292
Narenji, M., Talaee, M. R., & Moghimi, H. R. (2017). Effect of Charge on Separation of Liposomes upon Stagnation. Iranian journal of pharmaceutical research: IJPR, 16(2), 423–431.
C8. For clarity, the Figure 3 captions should indicate that the data are from the light scattering analysis.
Corrected in the main text.
Morphological characterisation of G-CSF-loaded liposomes
C9. Figure 4 purports to show transmission electron micrographs of liposomes. The confusing captions do not make clear what sort of liposomes images (a) and (b) are showing. Were unloaded liposomes similarly micrographed? The scale bars are not clearly visible, and it is not explained why two different scales are used. Both images appear to be of the same scale, if they represent the same sample.
In the context of our study, microphotographs of blank liposomes were also obtained, however we came up with the decision not to include them in the main text, as the study aimed to characterize G-CSF-loaded liposomes. Both images presented in the manuscript are contain G-CSF-loaded liposomes, but under different magnification. We clarified this issue in the main text. Additionally, we improved the visibility of the scale bars.
C10. The images show opaque particles, quite uncharacteristic of liposome electron micrographs. From these, it is difficult to tell if these par6cles are liposomes or material aggregates. It is certainly impossible to determine their lamellarity if these are indeed liposomes. Furthermore, the authors even state in lines 151-153 that micelles and nanoemulsions were also formed. How do the authors distinguish between the different types of nanopar6cle and how do they account for their impact on the sample characterisations?
Our study employed transmission electron microscopy (TEM), and opaque particles are typical for this type of microscopy (Y. Wang et al. (2022); D.A. Kuznetsova et al. (2021)). We also analyzed over 20 images for each sample and concluded on liposomes’ morphology as stated in the main text of the manuscript.
TEM images of mPEG2k–DSPE liposomes (left) and CPC6k liposomes (right) (Y. Wang et al. (2022), doi:10.1039/D1MA01079D)
Micrographs collected by TEM technique for IA-16(OH)/DPPC modified liposomes at 0.029:1 (0.2 mM/7 mM) amphiphile/lipid molar ratio; 25℃ (D.A. Kuznetsova et al. (2021), doi:10.1016/j.colsurfa.2020.12567)
Regarding micelles and nanoemulsions, very few were formed, and their impact on the sample characteristics was not considered. Distinguishing between different types of nanoparticles in this case can be challenging, and in the future, we plan to pay more attention to this aspect using cryo-microscopy.
Determination of the granulocyte-colony stimulating factor extinction coefficients
Protein quantitation is typically performed using UV absorp6on at 280 nm, exploiting absorp6on by aromatic amino acid side groups. As the authors rightly pointed out, this is not the wavelength at which proteins maximally absorb. Instead, their absorption (and therefore the sensitivity of their detection) would be higher at 210 nm instead, where peptide bonds are involved.
C11. It is therefore unclear why quantitation at both 210 nm and 280 nm were performed. In lines 115-117, the authors even assert that quan6ta6on at only one wavelength is desirable.
Thank you for comment. The main text of the manuscript reports «While determining the encapsulation efficiency of G-CSF, our study used a wave-length of 210 nm, as the greatest increase in optical density was recorded at this wave-length. This is probably explained by the fact that the content of aromatic amino acids in the structure of G-CSF is less than 10% [33] making the wavelength of 280 nm the least suitable for these purposes».
Preliminary data on extinction at different wavelengths were provided solely to justify our choice of method, and the application of a single wavelength minimizes potential errors and variations associated with averaging data obtained from two different wavelengths.
C12. They set about generating linear regressions of absorption from G-CSF at varying concentrations using both 210 nm and 280 nm. These alone, would have allowed for quantitation of samples. However, the authors also determined the extinction coefficient of G-CSF at these wavelengths for reasons that are not given. Granted, if quantitation of G-CSF at 210 nm is novel, then reporting its extinction coefficient at this wavelength would be useful information. Whether this is so should be made clear. Otherwise, the exercise seems rather arbitrary. Their method of deriving the extinction coefficient from the linear regression is also not explained.
C13. To determine the G-CSF extinction coefficient, the authors used the Bouguer-Lambert- Beer law expressed as Equation 1. The equation given uses optical density (D) instead of absorbance for the calculation. This law uses absorbance, which is also the unit indicated in Figure 1, not optical density. The absolute values may be the same, but the meaning is not.
Thank you for your comment. In our study, we employed a UV-spectrophotometric method to determine the encapsulation efficiency of G-CSF, which requires calculating the molar extinction coefficient. This step was necessary to utilize the Beer-Lambert law, allowing for quantitative assessment of protein content in solution based on its optical density. Since information regarding the purity of G-CSF was provided and confirmed solely by the manufacturer, we decided to calculate the molar extinction coefficient directly from our data to verify that the UV-spectrophotometry is suitable for accurate determination of G-CSF concentration.
Determining the extinction coefficient also allows for consideration of any potential contaminants in the G-CSF preparation and ensures that the absorption measurements are based on the concentration of pure protein. Thus, the calculation of the extinction coefficient was a crucial step in ensuring the reliability of the quantitative assessment of the samples.
Determination of the granulocyte-colony stimulating factor encapsulation efficiency
The authors go on to measure the encapsula6on efficiency of G-CSF in the liposomes, as well as the loaded and unloaded liposome average size, polydispersity, and surface charge.
C14. Considerable emphasis was placed by the authors on the encapsulation efficiency achieved. The authors used their linear regressions to determine this value. As an indicator of the quality of their standard curves, the authors report the R2 values obtained. The R2 values are wrongly described as correlation coefficients (line 107). Correla6on coefficients are R not R2 values. R2 values are coefficients of determination. These two coefficients indicate very different things.
Thank you for your comment. We acknowledge the error and have already corrected it. The R² values are defined as coefficients of determination, rather than correlation coefficients. Thank you for your careful reading!
C15. However, the linear regressions appear to have been generated by single readings, since none of the points show uncertainty values. In fact, only the determinations for polydispersity and surface charge were explicitly reported to have been performed in triplicate (Lines 364-367).
Thank you for your comment. We would like to clarify that a calibration curve was developed before each new experiment to ensure the accuracy. The graph provided in the article is representative and intended to familiarize readers with a typical result. All measurements were performed with necessary replicates to achieve high reliability of the results.
They reportedly determined an encapsulation efficiency of 63.1 ± 4.7 %. However, they used both 210 nm and 280 nm to make this determina6on. It is difficult to accept that the values obtained from two different wavelengths would agree with such precision. If this value is the result from a single wavelength, then the source of the uncertainty given is unclear since the degree of replication of both the standard curve determina6on as well as the sample analysis is not given.
Even if it is assumed that 210 nm was used, the accuracy of the encapsulation efficiency reported is in doubt, as explained below.
As previously mentioned, we decided to use a wavelength of 210 nm to assess the encapsulation efficiency. This choice is based on the observation that the highest increase in optical density was recorded at this wavelength, making it more suitable for our purposes. Preliminary data obtained at other wavelengths were only used to justify this choice, but the main encapsulation efficiency was calculated exclusively at 210 nm.
C16. Free G-CSF was purported separated from the G-CSF-loaded liposomes using sizeexclusion chromatography. They do not explain how the fraction containing the G-CSF-loaded liposomes was collected or identified.
We collected several fractions of 750 µL each and analyzed them using dynamic light scattering (DLS) to confirm the presence or absence of liposomes. Additionally, visual opalescence of the liposomes-containing samples determined the indeed presence of liposomes. We subsequently pooled the aqueous phase into a single volume and analyzed it by spectrophotometry to evaluate the content of free G-CSF. The column was also washed with water, and this fraction (the washout) was separately analyzed for substance content by spectrophotometry.
C17. To extract the encapsulated G-CSF from the loaded liposomes, the authors had disrupted the liposomes using ethanol. There is no mention of purifying the G-CSF prior to absorption quantitation. In other words, the sample had contained ethanol and soy PC in addition to the G-CSF. Both ethanol (UV cutoff = 210 nm) as well as soy PC would have absorbed at 210 nm as well. As such, the absorp6on the authors measured would have been due to more than just the G-CSF, meaning that they have over-estimated their encapsulation efficiency.
Thank you for your comment. We would like to note that we conducted additional studies aimed at assessing the effects of various solvents used for the liposomes’ disruption (ethanol, methanol, SDS, Triton X-100), as well as the liposomes themselves, on their spectral characteristics in the UV-visible spectrum. The results indicated that their influence on the spectrum at 210 nm is insignificant and can be considered during the experiment. Thus, the likelihood of overestimating the encapsulation efficiency was minimized.
Long-term storage stability of G-CSF-loaded liposomes
The ability to preserve a drug formulation over long periods is distinctly desirable, in order to ensure adequate long-term drug supplies. As such, the authors’ investigation into the longterm storage stability of their G-CSF-loaded liposomes is an important one.
C18. However, while the authors show that the G-CSF-loaded liposomes retain their polydispersity, size, and surface charge for up to 60 days at 4 °C, they do not report on the state of the G-CSF inside. It is also important that the drug, and not just the drug carrier, remains stable. The authors do not present any stability nor func6onality data for the encapsulated G-CSF.
You are absolutely correct, the evaluation of encapsulation efficiency (%EE) plays a crucial role in confirming the stability of liposomal formulations. In addition to the aforementioned aspects, it is worth noting that the encapsulation efficiency of G-CSF was measured also at 3-month storage (data not presented in the main text), and during the analysis, no significant changes in encapsulation efficiency were observed. This indicates that the liposomes maintained their integrity and ability to retain the encapsulated substance during this period.
However, despite the positive results at 3 months period, it is important to conduct regular %EE assessments at all stages of storage for a more comprehensive confirmation of the long-term stability and retention of the encapsulated substance. Thus, in future studies, we plan to pay special attention to the dynamics of encapsulation efficiency at different time points to completely rule out the possibility of active substance loss.
C19. The purpose of Figure 5d is unclear.
Thank you for your comment. Since the figure raised additional questions, we decided to remove it from the final version of the manuscript. Its initial purpose was to demonstrate the lack of changes in the main characteristics of liposomal G-CSF over time. In order to prevent possible misunderstandings, it was excluded.
Effect of G-CSF-loaded liposomes on mitochondrial function / cytotoxicity
The final set of data presented is variously described as indicative of changes in mitochondrial function, and cytotoxicity. The ul6mate aim, to put it in simple language, is to see if the G-CSF-loaded liposomes can cause harm to cells (cytotoxicity).
They chose to use an assay that probes for different mitochondrial activities, including oxygen consumption, proton transport, and ATP production. From these data, the authors appear to assert that the liposomes are not cytotoxic.
There are a whole host of ques6ons that need to be answered before such a definitive conclusion can be reached:
C20. In line 409, the G-CSF-loaded liposomes were described as purifed of unencapsulated GCSF using centrifuga6on at 2,000 x g for 10 mins. Unlike size exclusion chromatography, centrifugation using such a low force is unlikely to cause nanoparticles to pellet. Typically, ultracentrifugation is needed to harvest liposomes, unless the G-CSF has increased their densities significantly. Did the authors confirm that the pellets contained G-CSF-loaded liposomes?
Thank you for your comment. Subsequently, we used centrifugation at 10,000 x g on different equipment (Gyrozen Mini Microcentrifuge, Gyrozen, Gimpo, South Korea) to prepare G-CSF-loaded liposomes in order to achieve better separation of liposomes from free G-CSF. The experimental results, including assessments of mitochondrial respiration, did not provide significant differences in parameters compared to liposomes purified at 2,000xg. Thus, we concluded that centrifugation at 2,000 x g in our case was sufficient to obtain G-CSF-loaded liposomes.
C21. How long were the cells actually exposed to the G-CSF-loaded liposomes prior to the analysis?
Lines 414-416 states that the cells were “……incubated for 24 h in a CO2 incubator (Sanyo Co., Ltd., MCO-19AIC, Osaka, Japan) to adapt the cells to the measurement conditions with the addition of test compounds at different concentrations……” From the language used, it is not clear if the measurement conditions included the test compounds. If the cells needed to adapt to the test conditions, then the test compounds should not have been included at this time.
But if they had not been included, then how long were the cells exposed to G-CSF-loaded liposomes?
Nonetheless, at best, the data represents mitochondrial function aner exposure for 26.3 hrs and at the worst, for 140 mins. How does this translate to a definitive conclusion of lack of cytotoxicity?
Thank you for your question and careful text reading. In our study, the standard incubation time for cells with test compounds was 24 hours, which is a generally accepted period for determining potential cytotoxicity. During this period, cells adapt to the experimental conditions, and it is possible to determine both the direct effect of the compound and possible cell death, if it occurs.
Regarding your question about mitochondrial inhibitors, this is part of the standard protocol for cellular respiration assay. At first, before inhibitors administration the cells are given time to adapt to basal conditions, and then oxygen consumption is measured. This information is presented in the “Materials and Methods” section, containing a detailed description of the experiment and the logic of its conduct. As there were no signs of cell death and significant cytotoxicity after 24 hours of exposure to the tested compounds, we concluded on the safety of liposomal G-CSF.
C22. It is hard to follow their analyses for mitochondrial function. For instance, how did the authors come to the conclusion that “This indicates a temporary nature of the tested prepara6ons’ effect and the absence of significant morphological changes within cells” (Lines 201-203) or that “no effect was observed on reserve respiratory capacity” (which there appears to be, from the different values of Maximal Respiration – Basal Respiration)?
Thank you for your comment. We confirm that a reliable difference is observed at basal respiration stage (4.30±0.07 for controls vs 5.84±0.12 for 10 mg/ml, p = 0.000011) and after oligomycin introduction (2.47±0.09 for controls vs 4.32±0.1 for 10 mg/ml, p = 0.000003), which inhibits ATP synthesis. The absence of reliable difference at subsequent stages, such as maximal respiration and respiratory chain inhibition, indicates a lack of significant proton leakage. This confirms the reversible nature of the effect in the context of cellular respiration leading to a conclusion that there are no significant cell morphological changes.
C23. When adding G-CSF-loaded liposomes to the cells, the authors describe addition of test compounds at different concentrations (1.25; 2.5; 5; 10 mg/mL) (Lines 416-417). Are these concentrations referring to phospholipid content, or G-CSF content? If they refer to G-CSF concentrations, why were different concentrations used for the unencapsulated G-CSF samples?
These concentrations refer to the G-CSF. Comparable concentrations were used for samples containing unencapsulated G-CSF after consideration of the G-CSF encapsulation degree, as well as other factors that may have arisen during G-CSF production, such as protein stability and variability of different batches. This ensured a correct comparison of liposomal and unencapsulated G-CSF for analysis of their effect in the experiment.
C24. The Figure 6 captions contain Cyrillic script that is not explained. (b) is assumed to refer to unencapsulated G-CSF, and (c) to unloaded liposomes.
Corrected.
C25. What is the significance of the “dose-dependent effect of G-CSF liposomes” (Lines 454-455)?
The concept of "dose-dependent effect of G-CSF liposomes" refers to the changes in cellular response with changing concentrations of G-CSF-loaded liposomes. Our study established a positive correlation between mitochondrial respiration (basal respiration and after oligomycin administration) and increasing concentrations of G-CSF-loaded liposomes. The obtained data are important for selection of optimal dosages and administration regimens, which can be beneficial for the future clinical application of liposomal of G-CSF.
C26. Were other cellular functions evaluated?
Current study aimed to evaluate the cell metabolic activity based on MTT analysis. However, cell morphology during the experiment was assessed by microscopy.
C27. Did the authors perform simple viability assays which track growth and reproduction? They explained (although not clearly) why the MTT assay was excluded, but what about other enzyme-based methods such as calcein-AM staining. Etc?
Our study preliminary assessed the safety of G-CSF-loaded liposomes in human dermal fibroblasts due to their versatility for in vitro safety evaluation. We recognize the importance of viability assays, including calcein-AM staining, and we plan to add this method in future research. In addition, we plan to conduct an extended analysis on different cell types for comprehensive characterization of the safety and efficacy of G-CSF-loaded liposomes.
C28. Did the cells continue to live and grow aner exposure to G-CSF-loaded liposomes?
During the experiments in Seahorse XFe96 Real-Time Cell Metabolic Analyzer, the administration of rotenone and antimycin A resulted in cell death due to mitochondrial respiration inhibition. Nevertheless, under visual control cells remained viable after treatment with GSF-loaded liposomes before the mentioned inhibitors administration.
C29. How did the G-CSF-loaded liposomes actually interact with the cells?
We did not evaluate this aspect in the framework of our research. Our work focused on assessing the stability and safety of G-CSF-loaded liposomes, while a detailed study of the interaction between liposomes and cells is planned for future research.
C30. Did the liposomes release the G-CSF outside the cells? If so, how and why?
Current in vitro studies demonstrate the release of more than 70% of G-CSF within 24 hours from liposomes. These findings were not included in present manuscript. However, according to mitochondrial function assessment, no cytotoxicity was observed when using liposomes, as well as unencapsulated G-CSF, which may indirectly indicate the prolonged action of liposomal G-CSF.
C31. Were the G-CSF-loaded liposomes taken up by the cells? If so, how?
C32. If they were endocytosed, how did they escape endosomal entrapment? Aner that, how did they release their G-CSF?
C33. How long did the liposomes persist inside or outside of the cells?
These aspects were not evaluated in present research (C31, C32, C33). Our work was focused on assessing the stability and safety of G-CSF-loaded liposomes, while a detailed study of the interaction between liposomes and cells is planned for future experiments.
C34. What would the effect of G-CSF-loaded liposomes be on other cell types? E.g. blood cells, etc? The authors do men6on this need in lines 320-322. However, this undermines their definitive judgement of their G-CSF-loaded liposomes as being safe.
These, and many more ques6ons need to be addressed before the safety of the liposomes can be reasonably assured, much less for progress to in vivo testing to be justified.
Our study preliminary assessed the safety of G-CSF-loaded liposomes in human dermal fibroblasts due to their versatility for in vitro safety evaluation. We recognize the importance of viability assays, including calcein-AM staining, and we plan to add this method in future research. In addition, we plan to conduct an extended analysis on different cell types for comprehensive characterization of the safety and efficacy of G-CSF-loaded liposomes.
- Presentation
The manuscript appears to have been written separately by at least three people with distinct styles. One is particularly proficient linguistically, while the other two, who appear to have taken care of the technical sections, appear to have difficulties with English. Table and Figure references and cap6ons have mistakes which hinder understanding.
D1. Many company names in the Materials section are not properly or correctly given.
Some company names given in the manuscript are located in Russian Federation, explaining difficulties in perception.
D2. Table 1 describes “phosphate buffer solution”. This should more correctly be described as “phosphate-buffered saline” as is done everywhere else.
Corrected.
D3. Figure 2 includes an unnecessary elongated dash before the wavelengths, leading to possible confusion.
Corrected.
D4. The Figure 4 caption is confused, as explained above.
Corrected.
D5. Line 199 refers to Figure 6A which does not exist.
Corrected.
D6. Elsewhere are odd phrases like those pointed out above, or wrong words (e.g. line 229: “is attributed to” instead of “contributes to”, line 236: “interest to liposomes” instead of “interest in liposomes”, line 255: “dispersity” instead of “polydispersity”, etc) that need correcting. I would advise having the entire manuscript vetted by the English-proficient author, especially the technical sections. If possible, keep to one style of wri6ng for easy reading.
The manuscript underwent extensive English language editing.
Conclusions
I would like to thank the authors for their work. However, while the topic is an interesting and indeed important one, it is with regret that I cannot recommend this work for publication.
There are severe scientific and technical problems that need to be addressed before the manuscript can be adequate. It is also difficult to agree that the findings actually contribute significantly to the existing body of knowledge, if at all. This is something illustrated by the fact that the report’s generic title describes work that has already been reported as early as 20 years ago (as cited in doi: 10.1006/bbrc.1994.1247.).
The scientific novelty of our study is demonstrated in B3.
Nonetheless, I hope the authors will not be discouraged by this feedback, and will instead use it as a guide to improve their research. I look forward to the possibility of evaluating their work positively next time.
Reviewer 3 Report
Comments and Suggestions for Authors
The manuscript describes a well-structured study on the development and evaluation of liposome-based drug delivery systems, specifically for encapsulating Granulocyte Colony-Stimulating Factor (G-CSF). The researchers successfully created G-CSF-loaded liposomes characterized by impressive stability and encapsulation efficiency. In vitro analysis indicated no cytotoxicity and showed dose-dependent viability effects on human dermal fibroblasts, suggesting potential for safe therapeutic applications. However, there are several issues that require further clarification.
1. In “Preparation of granulocyte colony-stimulating factor-loaded liposomes,” the description mentions the use of phosphatidylcholine, but it does not specify the source type (e.g., soybean, egg) or any other lipids that might be present. It is important to include the full lipid composition, including any cholesterol or other additives, to ensure reproducibility.
2. In the method for “Determination of the encapsulation efficiency (EE)”, the use of spectrophotometry alone for determining EE may be prone to interference, particularly from residual solvents or drugs adsorbed onto the liposome surface. To enhance the accuracy of your results, I recommend considering additional methods for determining EE. High-Performance Liquid Chromatography (HPLC) is a powerful technique that could complement the spectrophotometric data, providing a more comprehensive understanding of the EE.
3. In the assessment of the effect of G-CSF-loaded liposomes, there is currently no direct evidence or data available to confirm that G-CSF has been successfully delivered into the cells. To address this gap, it is recommended to perform experimental assessments of the cellular uptake of G-CSF-loaded liposomes. This can be effectively achieved by using fluorescence-labeled liposomes (since G-CSF may not inherently possess fluorescence) and employing advanced techniques such as fluorescence microscopy or flow cytometry.
4. To strengthen the universality and applicability of the findings, it is recommended to expand the functional assay to test on different cell lines.
5. The legend of figure 6 contains several non-English words; please provide a clearer statement in English.
Comments on the Quality of English LanguageThe quality of the English language is generally good, with precise and technical language suitable for a scientific manuscript. However, there are still some minor errors, for example: the legend of Figure 6 contains several non-English words; please provide a clearer statement in English.
Author Response
The manuscript describes a well-structured study on the development and evaluation of liposome-based drug delivery systems, specifically for encapsulating Granulocyte Colony-Stimulating Factor (G-CSF). The researchers successfully created G-CSF-loaded liposomes characterized by impressive stability and encapsulation efficiency. In vitro analysis indicated no cytotoxicity and showed dose-dependent viability effects on human dermal fibroblasts, suggesting potential for safe therapeutic applications. However, there are several issues that require further clarification.
- In “Preparation of granulocyte colony-stimulating factor-loaded liposomes,” the description mentions the use of phosphatidylcholine, but it does not specify the source type (e.g., soybean, egg) or any other lipids that might be present. It is important to include the full lipid composition, including any cholesterol or other additives, to ensure reproducibility.
Dear reviewer, In response to your inquiry, we would like to clarify that section 4.1 "Materials" contains specified data on the commercially available phosphatidylcholine that we used, including manufacturer’s information regarding its type and fatty acid composition. In particular, we utilized soybean phosphatidylcholine Lipoid S-100 (PC, Lipoid, Ludwigshafen, Germany, CAS: 97281-47-5). The fatty acid composition of this phosphatidylcholine is as follows: palmitic acid — 15%, stearic acid — 3%, oleic acid — 12%, linoleic acid — 62%, and alpha-linolenic acid — 5%.
Soy phosphatidylcholine is a natural phospholipid that is widely used in liposomes’ development due to its availability and formation of stable membranes. In our study, we did not incorporate cholesterol, as this could reduce the encapsulation efficiency of G-CSF due to increased rigidity of the lipid bilayer, a finding supported by V.A. Shchelkonogov et al. (2021).
Shchelkonogov, V.A., Darnotuk, E.S., Chekanov, A.V. et al. Liposomal Drug with Carnosine
and Lipoic Acid: Preparation, Antioxidant and Antiplatelet Properties. BIOPHYSICS 66,
278–285 (2021). https://doi.org/10.1134/S0006350921020214
- In the method for “Determination of the encapsulation efficiency (EE)”, the use of spectrophotometry alone for determining EE may be prone to interference, particularly from residual solvents or drugs adsorbed onto the liposome surface. To enhance the accuracy of your results, I recommend considering additional methods for determining EE. High-Performance Liquid Chromatography (HPLC) is a powerful technique that could complement the spectrophotometric data, providing a more comprehensive understanding of the EE.
Our study used a spectrophotometric method to determine the encapsulation efficiency (EE) due to its simplicity and cost-effectiveness, making it easily applicable for quantitative assessment. According to the research by Qureshi H. et al. [49], this method demonstrates sufficient accuracy, sensitivity and specificity, and complies with ICH (International Council on Harmonisation) recommendations, making it suitable for routine G-CSF analysis.
However, we acknowledge that employing additional methods, such as high-performance liquid chromatography (HPLC), could enhance the accuracy of EE assessment and provide a more comprehensive understanding of the encapsulation processes. Future studies plan to integrate HPLC to obtain more precise data on encapsulation efficiency.
We would also like to note that we conducted additional studies to evaluate the impact of various solvents used for liposome disruption (ethanol, methanol, SDS, Triton X-100) and the liposomes themselves on spectral characteristics in the UV-visible range. As a result, the influence turned out being insignificant, thus, it can be considered during the experiment.
Thank you once again for your recommendation, we believe that further research will improve our results.
- In the assessment of the effect of G-CSF-loaded liposomes, there is currently no direct evidence or data available to confirm that G-CSF has been successfully delivered into the cells. To address this gap, it is recommended to perform experimental assessments of the cellular uptake of G-CSF-loaded liposomes. This can be effectively achieved by using fluorescence-labeled liposomes (since G-CSF may not inherently possess fluorescence) and employing advanced techniques such as fluorescence microscopy or flow cytometry.
Thank you for valuable comment. The primary goal of this study was to evaluate the stability and safety of the developed G-CSF loaded liposomes. We focused on the physicochemical characteristics and biological safety of the liposomal formulations, which is a crucial step prior to further investigations.
We acknowledge that the cellular uptake assessment is an important aspect for a complete understanding of the active substance delivery effectiveness. In future research we plan to entail experiments for cellular uptake of G-CSF-loaded liposomes evaluation. However, we would like to note that the use of fluorescence-labeled liposomes requires further investigation, as the introduction of labels may affect the loading capacity of the liposomes with the active substance and lead to the results distortion in both false positive and false negative ways. This must be taken into account when interpreting such experiments, which we plan to explore in future studies.
- To strengthen the universality and applicability of the findings, it is recommended to expand the functional assay to test on different cell lines.
We are totally agree that such approach could significantly enhance the versatility and application potential of our results. However, the goal of our current study was a preliminary assessment of the developed G-CSF loaded liposomes on human dermal fibroblast. This choice was based on the fact that fibroblasts are an accessible and versatile cells for in vitro safety investigation of various pharmaceutical agents. Their use provides a reliable and representative data for the initial phase of research, which is important for the preliminary assessment of the stability and safety of new drug formulations.
We recognize the proposed approach as a topic for future research, in which we plan to conduct an extensive functional analysis on various cell types. This will allow us to more thoroughly characterize the safety and efficacy of G-CSF-loaded liposomes and enhance the versatility of our findings.
- The legend of figure 6 contains several non-English words; please provide a clearer statement in English.
Corrected, we are sorry for unclear statement.
Round 2
Reviewer 2 Report
Comments and Suggestions for Authors
The reviewer thanks the authors for taking the time to address the problems raised, and hopes that they would consider these new comments carefully.

Please see the uploaded document for comments.
Author Response
Second Review of “Formulation and in vitro characterization of granulocyte colony-stimulating factor loaded liposomes”, authored by Kseniia V. Obedkova, Galina Kh. Safarian, Dmitry V. Tovpeko, Julian R. Ryzhov, Olesya N. Bespalova, and Natalya I. Tapilskaya.
Overview
The reviewer thanks the authors for taking the time to carefully address the problems raised.
The three major issues identified in the first review are:
- Incorrect technical descriptions
- Missing explanations
- Problematic arguments and conclusions
It is not clear what the significance of the red text in the revised manuscript is. In some cases, the red text does not differ from the original, so the colour does not seem to indicate that an amendment has been made.
The red text highlights the revisions made by the authors, as well as edits performed by certified native English speaker after professional editing.
The following are the reviewer’s responses to the authors’ replies.
- Aims
Response:
The introduction prepares the reader for what is to come. To expand on the authors’ explanation, the introduction not only provides a thorough description in the field of current research, it also needs to clearly identify the problem that needs addressing. This would make it easier to understand why the authors have selected their specific aims. As the authors have indicated, a brief mention of the aims is enough, while a detailed explanation can be reserved for the Discussion.
The problem is precisely that the authors have not mentioned the main goal of the study. In a single line, it is stated that the aim is “to obtain a liposomal drug formulation of G-CSF and to evaluate its chemical and biological characteristics, including physicochemical stability and safety”.
With all due respect to the authors, this was not their actual aim at all. The aims were, broadly speaking:
(i) To produce G-CSF liposomes with a specific size range, low polydispersity, high encapsulation efficiency, low toxicity, long-term physicochemical stability, etc;
(ii) To demonstrate using a variety of assays that this was successful;
(iii) To do this in order to demonstrate how G-CSF liposomes should be produced, to make them acceptable for clinical use.
These points are scattered throughout the manuscript, but would have made the authors’ aims crystal clear had it been included immediately in the introduction. This would also have allowed the authors to prepare the reader for the methods used in order to achieve this. Which brings the discussion to the mention of methods in the introduction.
The authors explained that they had intended to finish the introduction “without delving into the specifics of the applied methods” and that “the methods are described in greater detail in the "Materials and Methods" section”. The applied methods were not mentioned at all in the introduction, in fact.
The suggestion by the reviewer was not to provide the methods (detailed description), but the methodology (strategy). The methodology would have explained why the authors had selected specific assays for the task, even if the assays themselves were not mentioned in the introduction.
In fact, this is nicely explained in lines 255 – 279 of the revised manuscript.
Finally, a clear presentation of the aims, and a succinct explanation of the parameters selected for investigation, would have clearly shown the novelty of the authors’ work. This is addressed in greater detail in the next section.
The aim and novelty of current study was to obtain G-CSF-loaded liposomes sized less than 200 nm with low polydispersity (below 0.3) and high encapsulation efficiency, providing their long-term stability. The size of liposomes was selected based on literature data that liposomes less than 200 nm interact with cell membranes while avoiding complement activation and subsequent severe toxicity [Danaei M. et al. (2018); Pisani S. et al. (2023)]. Furthermore, the polydispersity index (PDI) was kept below 0.3, which is indicative of a uniform size distribution of liposomes.
We also conducted an initial in vitro assessment of cytotoxicity of the developed liposomal formulations. Since one of the well-known mechanisms of cytotoxicity involves mitochondrial damage, we performed in vitro mitochondrial function investigation on human dermal fibroblasts. Achieving the aforementioned characteristics of liposomes will further facilitate their application to clinical practice, ensuring increased safety and efficacy of liposomal formulations.
Danaei, M.; Dehghankhold, M.; Ataei, S.; Hasanzadeh Davarani, F.; Javanmard, R.; Dokhani, A.; Khorasani, S.; Mozafari, M.R. Impact of Particle Size and Polydispersity Index on the Clinical Applications of Lipidic Nanocarrier Systems. Pharmaceutics 2018, 10, 57. https://doi.org/10.3390/pharmaceutics10020057
Pisani, S.; Di Martino, D.; Cerri, S.; Genta, I.; Dorati, R.; Bertino, G.; Benazzo, M.; Conti, B. Investigation and Comparison of Active and Passive Encapsulation Methods for Loading Proteins into Liposomes. Int. J. Mol. Sci. 2023, 24, 13542. https://doi.org/10.3390/ijms241713542
While the authors repeatedly emphasised the encapsulation efficiency of their protocol, describing it as high, it is by no means remarkable when compared to those reported by the literature that they themselves cite. We agree with your observation that the encapsulation efficiency in our study is not outstanding in comparison to those reported in the literature. However, it is important to note that other studies (J. Meyer et al. (1994); F. Kiafar et al. (2016); S. Lewicki et al. (2020)) focused on such parameters as polydispersity (PDI) and particle size in addition to encapsulation efficiency. For instance, J. Meyer et al. (1994), demonstrated the ability to encapsulate from 2% to 90% of G-CSF. However, significant differences in diameter of liposomes and their heterogeneity were also noted: "The mean diameter of liposomes prepared using method 1 was 253 nm (polydispersity=0.2). The mean diameter of liposomes prepared using method 2 was 343 nm (polydispersity=0.6). The mean diameters and polydispersities of lyposomes prepared using method 3 and 4 were 765 nm (0.5) and 783 nm (0.4), respectively." Similar results were reported by F. Kiafar et al. (2016) and S. Lewicki et al. (2020), as the particles’ size and heterogeneity increased with higher degrees of encapsulation.
Response:
The authors basically agree that the encapsulation efficiency they obtained was not remarkable, yet continue to assert that it is remarkable in the report. The authors’ decision to focus on other parameters such as PDI and particle size still does not make their reported encapsulation efficiency any more outstanding. The mention of the other parameters, seem like non sequiturs.
The reviewer suspects, however, that the authors were trying to assert novelty here.
While high encapsulation efficiency is indeed important, however, if achieved by increased particle size, the clinical benefits of the liposomal formulation may be negated [Danaei M. et al. (2018)]. The significance of other parameters, such as particle size and PDI, relies in the fact that under clinical conditions larger liposomes may not penetrate tissues effectively, activate the immune system or cause toxicity. Therefore, it is crucial to attain high encapsulation efficiency and maintain parameters that ensure drug safety and effectiveness. Merely mentioning one characteristic without considering the others leads to a dead end.
As previously mentioned, other researchers in the field reported on larger sized G-SCF-loaded liposomes than ours. This emphasizes the importance of both encapsulation efficiency and control over particle size and PDI, which are critical for biodistribution and side effect reduction in clinical practice. Larger liposomes, even with high encapsulation efficiency, may prove ineffective or even hazardous if factors related to their interaction with cells and the immune system are not considered.
Danaei, M.; Dehghankhold, M.; Ataei, S.; Hasanzadeh Davarani, F.; Javanmard, R.; Dokhani, A.; Khorasani, S.; Mozafari, M.R. Impact of Particle Size and Polydispersity Index on the Clinical Applications of Lipidic Nanocarrier Systems. Pharmaceutics 2018, 10, 57. https://doi.org/10.3390/pharmaceutics10020057
In our study, the application of additional steps, including freeze-thaw cycles, resulted in the achievement of homogeneous liposomes up to 200 nm and a PDI < 0.3. This indicates that our liposomes, although having a more moderate encapsulation efficiency, are smaller in size and are highly homogenous, which are important factors for the stability and bioavailability of liposomal formulations.
Response:
Again, this explanation would have made it clear what lack in the literature the authors were addressing. Although the methods used are not novel, they did allow the authors to meet their aims as described above (but not in the manuscript).
The “Discussion” section contains: «…According to the results of our study, G-CSF-loaded liposomes of regular round, oval, or droplet shapes were obtained with an average particle size of 161.9±9.9 nm. They were homogeneous, had a low PDI value (less than 0.3), and were characterized by a high encapsulation efficiency of the active substance—63.1±4.7% (Table 2). The obtained G-CSF-loaded liposomes were stable for at least 2 months of storage at a temperature of 4 °C and relative humidity of 60%. This indicates the successful encapsulation of G-CSF within the liposomes and demonstrates the effectiveness of additional steps (intensive shaking and five repeated cycles of freezing–thawing) during their production. In addition, the excess of PC (G-CSF ratio 1:100 by weight) could have influenced the relatively high-efficiency encapsulation of G-CSF within the liposomes…».
B2. Furthermore, it is curious that the authors point out in lines 249-250 that gentle hydration usually results in low encapsulation efficiencies. We agree that gentle hydration generally leads to low encapsulation efficiency, especially under standard conditions. However, we chose this method as it preserves the biological activity of GCSF while minimizing the risk of protein denaturation, which is critically important for maintaining its therapeutic properties. To enhance encapsulation efficiency, we added freeze-thaw cycles. This approach was supported by previous studies (e.g., J.P. Colletier et al. (2002) and A.C. Khayrani et al. (2024)), which demonstrated significant improvement of encapsulation efficiency without substantial protein damage.
Thus, the application of gentle hydration was dictated by the need to preserve the functional activity of G-CSF, and the introduction of additional steps allowed increasing encapsulation efficiency to acceptable levels. The novelty of our study lies in the production of homogeneous G-CSF-loaded liposomes sized up to 200 nm and with PDI <0.3, as well as a detailed assessment of their stability over time, which is an important step toward the development of effective liposomal formulations. Our approach demonstrates that high homogeneity of liposomes and stable parameters can be achieved when encapsulating biologically active molecules such as GCSF, bringing these systems closer to clinical application.
Response:
Once again, here is a really nice summary of the authors’ aims and methodology. Including this information would have vastly clarified the report.
The information was added.
Nevertheless, we acknowledge that the methods we used to prepare liposomes are based on well-known and established techniques developed over decades, such as the lipid film hydration method and freeze-thaw cycles. Despite their widespread application, the combination of these methods under optimized conditions allowed us to achieve liposomal characteristics not previously reported in other studies.
Colletier, J. P., Chaize, B., Winterhalter, M., & Fournier, D. (2002). Protein encapsulation in liposomes: efficiency depends on interactions between protein and phospholipid bilayer. BMC biotechnology, 2, 9. https://doi.org/10.1186/1472-6750-2-9
Khayrani, A.C., Fahmi, M., Nurhayati, R.W., Manas, N.H.A., Suhaeri, M., 2024. Effect of Freeze- Thaw Cycles Method to Transfersome Characteristics for Growth Protein Encapsulation. International Journal of Technology. Volume 15(2), pp.267-278
Response:
And here is another useful explanation of the aims and methodology. The reviewer thanks the authors for the references provided.
The information was added.
B3. Lines 289-291 appear to suggest that the works cited do not provide (detailed) information on long-term storage stability of G-CSF-loaded liposomes. A study conducted by J. Meyer et al. (1994) primarily focused on the technology of G-CSF encapsulation in liposomes and evaluated the active substance’s release both in vitro and in vivo. However, authors did not provide detailed information regarding the long-term stability of liposomes during storage and their biocompatibility. The study by F. Kiafar et al. (2016) proposed a prediction model on characteristics of filgrastim (G-CSF)-loaded liposomes. This model was used to create optimal liposome formulations with improved properties, such as size and encapsulation efficiency. However, it is important to note that the evaluation of liposomes’ biological properties, as well as their long-term stability was not conducted. Only S. Lewicki et al. (2020) performed a comprehensive analysis of G CSF-loaded liposomes, including the assessment of their biological activity and stability during storage over a period of 28 days under various conditions. However, the described liposomes significantly differ from ours in particle size.
Response:
And another example. The reviewer’s point must surely be clear to the authors by now.
The information was added.
Additional comment:
The authors focused on the stability of the liposomes in their work. However, it should not be forgotten that the function of the liposomes is also to preserve the G CSF functionality. As such, while the stability of the liposomes is important, it is a moot point if the retained G-CSF is not functional. Ascertaining whether the load had retained its function, especially after long-term storage, should have been part of the characterisation efforts.
We agree that the functional properties of G-CSF should be evaluated after prolonged storage. However, questions regarding the preservation of its activity require further investigation and are beyond the scope of the current study. We currently conduct relevant experiments to assess how well G-CSF retains its activity after encapsulation and long-term storage.
It is noteworthy that when evaluating various sample preparation methods using centrifugation at 10,000 x g and 2,000 x g, the experimental results, including assessments of mitochondrial respiration, showed no significant differences in the indicators compared to liposomes purified at 2,000 x g, both immediately after preparation (the next day) and after prolonged storage. This study was conducted after three months of storage, which indirectly supports the preservation of functional activity. However, these data were not included in the main text of the article as they fall outside the current investigation.
- Methods and Conclusions
C5. Lines 354-356 explain that unilamellar vesicles were obtained by filtration through a syringe filter.
We concur with your observation that filtration through a syringe filter alone is not always sufficient to obtain unilamellar liposomes. However, our study additionally utilized sonication, which, according to the literature, promotes unilamellarity (R. Silva et al. (2010); J.-S. Yun et al. (2023); C.F. de Freitas et al. (2019)). Nevertheless, further research is required to fully confirm this assertion, such as the application of cryo-microscopy and the selection of appropriate conditions for sonication.
Silva, R., Ferreira, H., Little, C., & Cavaco-Paulo, A. (2010). Effect of ultrasound parameters for unilamellar liposome preparation. Ultrasonics sonochemistry, 17(3), 628–632. https://doi.org/10.1016/j.ultsonch.2009.10.010
Yun, J.-S.; Hwangbo, S.-A.; Jeong, Y.-G. Preparation of Uniform Nano Liposomes Using Focused Ultrasonic Technology. Nanomaterials 2023, 13, 2618. https://doi.org/10.3390/nano13192618
de Freitas, C. F., Calori, I. R., Tessaro, A. L., Caetano, W., & Hioka, N. (2019). Rapid formation of Small Unilamellar Vesicles (SUV) through low-frequency sonication: An innovative approach. Colloids and surfaces. B, Biointerfaces, 181, 837–844. https://doi.org/10.1016/j.colsurfb.2019.06.027
Response:
The reviewer thanks the authors for the references, and is glad that they agree that more evidence is needed in order to assert unilamellarity. And yet, they continue to assert that their liposomes are unilamellar (lines 357-359). Despite agreeing with the reviewer, the authors also continue to assert that it was the filtration through a 0.22 μm syringe filter that had conferred unilamellarity (same lines).
Corrected.
C7. Lines 131-134 state that all the liposome formulations they’d produced were neutrally charged, and also explained that neutrally charged liposomes are desirable for use in systemic administration. We acknowledge that phrasing in lines 131-134 may raise questions, and we appreciate your attention to this aspect. However, the minor changes in zeta potential that were recorded can be disregarded, as studies (D.Y. Wang et al. (2020); M.C. Smith et al. (2017); Y. Nie et al. (2012); M. Campos-Martorell et al. (2016); M. Narenji et al. (2017)) indicate that liposomes with zeta potentials ranging from -10 to +10 mV are considered electrically neutral. Thus, it can be asserted that the obtained G-CSF-loaded liposomes remain neutral, similar to the blank ones.
Response:
The reviewer thanks the authors for the references. However, this appears to be an appeal to authority and is not a convincing argument. Even if these references had been cited in the manuscript, it does not explain why such values of surface charge can be disregarded. Perhaps it can be shown or explained that such a low range of values does not result in interaction with plasma proteins, etc. Some justification should be offered beyond merely citing precedence.
The authors also do not address the question of why loaded and unloaded liposomes have different surface charges.
According to a widely accepted fact based on data collected over the past 30 years, liposomes with zeta potential ranging from -10 to +10 mV are considered neutral. We did not aim to conduct a comprehensive review of all these studies, however, we referred to H. Aoki et al. (1997). Authors state: "The blood clearance of the liposomes with zeta potentials under +10 mV was comparable to that of the neutral liposomes." Notably, in their study, the neutral liposomes had a zeta potential of -4.3 ± 0.6 mV, which supports our approach to defining the neutrality of liposomes.
We also referenced the statement: "It is known that neutral liposomes have a longer circulation time and lower cytotoxicity compared to cationic liposomes, as they do not interact with negatively charged proteins in the serum and blood. Additionally, neutral liposomes exhibit delayed clearance in vitro compared to cationic liposomes [35]."
The clinical implications of parameters such as surface charge, size and PDI require further investigation, as different applications may impose various requirements on liposomes. In the "Introduction" we highlighted the wide range of G-CSF application, and the efficacy of our formulation will be evaluated in diverse therapeutic settings. However, the primary goal of this study was to create a stable and safe liposomal G-CSF, which is corroborated by the initial assessment of mitochondrial cytotoxicity on human dermal fibroblasts.
Regarding the charge differences between loaded and plain liposomes, as previously mentioned, both types have a neutral charge (within -10 to +10 mV). This should not significantly impact their clinical characteristics. The G-CSF loading may slightly alter the surface charge of liposomes, but, according to the literature, such changes within the neutral range do not significantly affect the formulation’s efficacy.
Aoki, H., Tottori, T., Sakurai, F., Fuji, K., & Miyajima, K. (1997). Effects of positive charge density on the liposomal surface on disposition kinetics of liposomes in rats. International Journal of Pharmaceutics, 156(2), 163–174 https://doi.org/10.1016/s0378-5173(97)00184-1
Morphological characterisation of G-CSF-loaded liposomes
C9. Figure 4 purports to show transmission electron micrographs of liposomes.
In the context of our study, microphotographs of blank liposomes were also obtained, however we came up with the decision not to include them in the main text, as the study aimed to characterize G-CSF-loaded liposomes. Both images presented in the manuscript are contain GCSF-loaded liposomes, but under different magnification. We clarified this issue in the main text.
Additionally, we improved the visibility of the scale bars.
Response:
It is not convincing that Figure 4b shows the liposomes at higher magnification. Apart from the smaller particles being more numerous in 4b, the average size of the larger particles look similar in both images. 4a and 4b look like images of similar samples at the same magnification.
Why include images at different magnifications at all?
Figures 4a and 4b indeed show G-CSF-loaded liposomes; however, they pertain to different batches of obtained liposomes. This explains the differences in the number of smaller particles in the images. We included images from two batches of liposomes to demonstrate the reproducibility of the preparation process and to illustrate possible variations in morphology and particle distribution between batches. The differing magnifications allowed for better visualization of these characteristics and confirmed the stability of liposomal properties despite the differences in production batches.
C10. The images show opaque particles, quite uncharacteristic of liposome electron micrographs.
Our study employed transmission electron microscopy (TEM), and opaque particles are typical for this type of microscopy (Y. Wang et al. (2022); D.A. Kuznetsova et al. (2021)). We also analyzed over 20 images for each sample and concluded on liposomes’ morphology as stated in the main text of the manuscript.
TEM images of mPEG2k–DSPE liposomes (left) and CPC6k liposomes (right) (Y. Wang et al. (2022), doi:10.1039/D1MA01079D)
Micrographs collected by TEM technique for IA-16(OH)/DPPC modified liposomes at 0.029:1 (0.2 mM/7 mM) amphiphile/lipid molar ratio; 25℃ (D.A. Kuznetsova et al. (2021), doi:10.1016/j.colsurfa.2020.12567)
Response:
The reviewer respectfully, but most emphatically, disagrees from experience. In fact, simply googling "TEM liposomes” would yield a plethora of examples that look vastly different from the images shown in Figure 4. Nonetheless, the authors provide two references to support their conclusion, which the reviewer thanks them for. The reviewer points out that doi:10.1016/j.colsurfa.2020.12567 does not identify any document at all. It is assumed that this is mistake. On the other hand, Wang et al. (doi:10.1039/D1MA01079D) provides micrographs of liposomes that are small and not of ideal quality. Nonetheless, the reviewer invites the authors to take a closer look. The images in Figure 6 of this report show particles which, although they have a dense core, also have a halo of less dense material around it. In fact, the CPC6k liposomes sometimes show a distinct membrane surrounding the dense core. Dense cores often present themselves when the liposomes encapsulate a cargo, such as a drug. This may, indeed, be so in the authors’ case. This is why the reviewer had asked what their unloaded liposomes looked like. More importantly, there is no evidence at all, in Figure 4, that the particles are vesicular in nature. The particles shown appear to be completely solid. As such, it is hard to accept the assertion that these are images of liposomes, let alone unilamellar liposomes.
10.1016/j.colsurfa.2020.125673
We do not rule out the fact that some of the protein could be distributed not only within the liposome but also in its membrane, which may have affected the structural perception in the images. Further research utilizing methods such as cryo-TEM and atomic force microscopy (AFM) are required to confirm this assumption. The mention of unilamellarity has been removed.
Regarding micelles and nanoemulsions, very few were formed, and their impact on the sample characteristics was not considered. Distinguishing between different types of nanoparticles in this case can be challenging, and in the future, we plan to pay more attention to this aspect using cryo-microscopy.
Response:
Non-liposomal particles are a possible contamination that needs to be addressed. Not probing for the presence of (or removing) micelles and nanoemulsions, as well as their effect on the data collected seriously undermines the authors’ claims of sample homogeneity, stability, safety, etc.
Our claims of sample homogeneity, stability and safety are not solely based on TEM results. Additional studies were conducted to confirm these characteristics, including particle size determination, polydispersity index (PDI), zeta potential (ZP), encapsulation efficiency, as well as assessment of mitochondrial cytotoxicity. These methods provide a more comprehensive understanding of the sample characteristics and allow for conclusions about its stability and safety without relying solely on morphological data.
Determination of the granulocyte-colony stimulating factor extinction coefficients
C11. It is therefore unclear why quantitation at both 210 nm and 280 nm were performed. Thank you for comment. The main text of the manuscript reports «While determining the encapsulation efficiency of G-CSF, our study used a wave-length of 210 nm, as the greatest increase in optical density was recorded at this wave-length. This is probably explained by the fact that the content of aromatic amino acids in the structure of G-CSF is less than 10% [33] making the wavelength of 280 nm the least suitable for these purposes».
Response:
The review does not ask why only 210 nm was used, but rather why data for both 210 nm and 280 nm were presented, in view of the statements above.
Since G-CSF contains aromatic amino acids, we decided to consider a 280 nm wavelength. However, due to their low abundance and less intense absorption, the focus shifted to 210 nm, which is sensitive to peptide bonds and yields more accurate results for the investigated protein. Nonetheless, both wavelengths (210 nm and 280 nm) are represented for completeness and in response to possible reviewer inquiries about the use of standard methodologies, as a wavelength of 280 nm is often employed for protein concentration determinations [Campardelli R. et al. (2016); Sanches B.C.P. et al. (2021)]. If the reviewer feels that the use of data at 280 nm is excessive, we can remove it.
Sanches BCP, Rocha CA, Martin Bedoya JG, da Silva VL, da Silva PB, Fusco-Almeida AM, Chorilli M, Contiero J, Crusca E, Marchetto R. Rhamnolipid-Based Liposomes as Promising Nano-Carriers for Enhancing the Antibacterial Activity of Peptides Derived from Bacterial Toxin-Antitoxin Systems. Int J Nanomedicine. 2021 Feb 10;16:925-939. https://doi.org/10.2147/IJN.S283400
Campardelli R., Espirito Santo I., Albuquerque E.C., De Melo S.V., Della Porta G., Reverchon E. Efficient encapsulation of proteins in submicro liposomes using a supercritical fluid assisted continuous process. J. Supercrit. Fluids. 2016;107:163–169. https://doi.org/10.1016/j.supflu.2015.09.007
Preliminary data on extinction at different wavelengths were provided solely to justify our choice of method, and the application of a single wavelength minimizes potential errors and variations associated with averaging data obtained from two different wavelengths.
Response:
And here is the actual explanation. The reviewer actually agrees that such data is important for justifying the decision to use absorbace spectrometry, and only at 210 nm. However, this clarification is still not provided in the report.
We provided an explanation of our choice for 210 nm wavelength application in the section 2.1.
«While determining the encapsulation efficiency of G-CSF, our study used a wave-length of 210 nm, as the greatest increase in optical density was recorded at this wave-length. This is probably explained by the fact that the content of aromatic amino acids in the structure of G-CSF is less than 10% [33], making the wavelength of 280 nm the least suitable for these purposes. Additionally, the application of a single wavelength minimizes potential errors and variations associated with averaging data from two wavelengths…».
C12. They set about generating linear regressions of absorption from G-CSF at varying concentrations using both 210 nm and 280 nm.
C13. To determine the G-CSF extinction coefficient, the authors used the Bouguer-Lambert- Beer law expressed as Equation 1.
C14. Considerable emphasis was placed by the authors on the encapsulation efficiency achieved.
Thank you for your comment. In our study, we employed a UV spectrophotometric method to determine the encapsulation efficiency of G-CSF, which requires calculating the molar extinction coefficient. This step was necessary to utilize the Beer-Lambert law, allowing for quantitative assessment of protein content in solution based on its optical density. Since information regarding the purity of G-CSF was provided and confirmed solely by the manufacturer, we decided to calculate the molar extinction coefficient directly from our data to verify that the UVspectrophotometry is suitable for accurate determination of G-CSF concentration.
Determining the extinction coefficient also allows for consideration of any potential contaminants in the G-CSF preparation and ensures that the absorption measurements are based on the concentration of pure protein. Thus, the calculation of the extinction coefficient was a crucial step in ensuring the reliability of the quantitative assessment of the samples.
Response:
The reviewer wishes to emphasise that the Bouguer-Lambert-Beer law uses absorbance, not optical density. Even if the authors wish to insist on using “optical density” in the text, they must surely see that this contradicts the axis names in Figures 1 and 2.
Herein lies a major confusion. Did the authors calculate the G-CSF extinction coefficient using the Bouguer-Lambert-Beer law (line 99)? Or determine it using linear regression (line 108)?
If the reviewer had to guess, it would be that the authors had calculated the extinction coeffient using the Bouguer-Lambert-Beer law, and then confirmed this value using linear regression, and that they had also used this comparison to ascertain the purity of their G-CSF. If this is correct, then such an explanation is sorely needed in the manuscript.
The Bouguer-Lambert-Beer law was employed to determine the extinction coefficient of G-CSF, which was utilized for calculating concentration based on absorption data. The linear association served for calibration curve construction, on the basis of which the encapsulation efficiency was determined.
After determining the extinction coefficient using the Bouguer-Lambert-Beer law, the encapsulation was calculated on the basis of concentration obtained by this law. Thus, our approach included both the calculation of the extinction coefficient and the use of the linear association to obtain accurate data on encapsulation.
The data on absorption has been corrected.
C15. However, the linear regressions appear to have been generated by single readings, since none of the points show uncertainty values. Thank you for your comment. We would like to clarify that a calibration curve was developed before each new experiment to ensure the accuracy. The graph provided in the article is representative and intended to familiarize readers with a typical result. All measurements were performed with necessary replicates to achieve high reliability of the results.
Response:
The reviewer assumes that “all measurements” includes the readings for each point of the calibration curves. If so, it is strange that no error bars are given, even if those provided had been exemplary graphs.
Corrected.
As previously mentioned, we decided to use a wavelength of 210 nm to assess the encapsulation efficiency. This choice is based on the observation that the highest increase in optical density was recorded at this wavelength, making it more suitable for our purposes. Preliminary data obtained at other wavelengths were only used to justify this choice, but the main encapsulation efficiency was calculated exclusively at 210 nm.
Response:
Once again, the question of number of replicates arises. How many times did the authors make the measurements to obtain the data for this determination?
The absorptions were determined in triplicate by UV-visible spectroscopy at minimum four five different dilutions.
The relevant information has been incorporated into the manuscript. The final result concerning the encapsulation efficiency has been revised.
C16. Free G-CSF was purported separated from the G-CSF-loaded liposomes using sizeexclusion chromatography.
We collected several fractions of 750 μL each and analyzed them using dynamic light scattering (DLS) to confirm the presence or absence of liposomes. Additionally, visual opalescence of the liposomes-containing samples determined the indeed presence of liposomes. We subsequently pooled the aqueous phase into a single volume and analyzed it by spectrophotometry to evaluate the content of free G-CSF. The column was also washed with water, and this fraction (the washout) was separately analyzed for substance content by spectrophotometry.
Response:
These are important details that should be included in the Methods.
The fractions’ volumes and other similar parameters depend on the initial data and conditions of the specific experiment. These data are part of standard routine procedures and it is not always appropriate to include them in the manuscript, as they do not significantly impact the key results of the study.
C17. To extract the encapsulated G-CSF from the loaded liposomes, the authors had disrupted the liposomes using ethanol.
Thank you for your comment. We would like to note that we conducted additional studies aimed at assessing the effects of various solvents used for the liposomes’ disruption (ethanol, methanol, SDS, Triton X-100), as well as the liposomes themselves, on their spectral characteristics in the UV-visible spectrum. The results indicated that their influence on the spectrum at 210 nm is insignificant and can be considered during the experiment. Thus, the likelihood of overestimating the encapsulation efficiency was minimized.
Response:
This evidence needs to be mentioned or, better yet, presented. Otherwise, it would appear to be a major oversight.
Standard procedures, such as assessing the impact of solvents on spectral characteristics, are conducted before each experiment to ensure the accuracy and reliability of the data. However, according to established practice, these procedures are typically not included in the main body of the article, as they are routine and do not significantly affect the key results of the study.
C18. However, while the authors show that the G-CSF-loaded liposomes retain their polydispersity, size, and surface charge for up to 60 days at 4 °C, they do not report on the state of the G-CSF inside.
You are absolutely correct, the evaluation of encapsulation efficiency (%EE) plays a crucial role in confirming the stability of liposomal formulations. In addition to the aforementioned aspects, it is worth noting that the encapsulation efficiency of G-CSF was measured also at 3-month storage (data not presented in the main text), and during the analysis, no significant changes in encapsulation efficiency were observed. This indicates that the liposomes maintained their integrity and ability to retain the encapsulated substance during this period. However, despite the positive results at 3 months period, it is important to conduct regular %EE assessments at all stages of storage for a more comprehensive confirmation of the long-term stability and retention of the encapsulated substance. Thus, in future studies, we plan to pay special attention to the dynamics of encapsulation efficiency at different time points to completely rule out the possibility of active substance loss.
Response:
The reviewer points out that the first review did not assert that “evaluation of encapsulation efficiency plays a crucial role in confirming the stability of liposomal formulations”. Nonetheless, the reviewer agrees with such an assertion.
However, the question was why the G-CSF itself was not evaluated. The authors appear to have missed the point.
We agree that the functional properties of G-CSF should be evaluated after prolonged storage. However, questions regarding the preservation of its activity require further investigation and are beyond the scope of the current study. We currently conduct relevant experiments to assess how well G-CSF retains its activity after encapsulation and long-term storage.
It is noteworthy that when evaluating various sample preparation methods using centrifugation at 10,000 x g and 2,000 x g, the experimental results, including assessments of mitochondrial respiration, showed no significant differences in the indicators compared to liposomes purified at 2,000 x g, both immediately after preparation (the next day) and after prolonged storage. This study was conducted after three months of storage, which indirectly supports the preservation of functional activity. However, these data were not included in the main text of the article as they fall outside the current investigation.
C20. In line 409, the G-CSF-loaded liposomes were described as purifed of unencapsulated GCSF using centrifugation at 2,000 x g for 10 mins. Thank you for your comment. Subsequently, we used centrifugation at 10,000 x g on different equipment (Gyrozen Mini Microcentrifuge, Gyrozen, Gimpo, South Korea) to prepare G-CSFloaded liposomes in order to achieve better separation of liposomes from free G-CSF. The experimental results, including assessments of mitochondrial respiration, did not provide significant differences in parameters compared to liposomes purified at 2,000xg. Thus, we concluded that centrifugation at 2,000 x g in our case was sufficient to obtain G-CSF-loaded liposomes.
Response:
The authors recognised the need to ascertain whether 2,000 x g was effective enough to pellet their liposomes, compared to 10,000 x g. Their conclusions were drawn from the fact that both forces produced similar data. However, this is indirect evidence that liposomes had been pelleted. Were the pelleted samples examined microscopically for the presence of the particles shown in Figure 4? This would have been more convincing direct evidence.
The authors do not agree with the reviewer. Microscopy also cannot serve as direct evidence. Microscopy results may vary depending on sample preparation, which also reduces their reliability as direct evidence.
C21. How long were the cells actually exposed to the G-CSF-loaded liposomes prior to the analysis? Thank you for your question and careful text reading. In our study, the standard incubation time for cells with test compounds was 24 hours, which is a generally accepted period for determining potential cytotoxicity. During this period, cells adapt to the experimental conditions, and it is possible to determine both the direct effect of the compound and possible cell death, if it occurs. Regarding your question about mitochondrial inhibitors, this is part of the standard protocol for cellular respiration assay. At first, before inhibitors administration the cells are given time to adapt to basal conditions, and then oxygen consumption is measured. This information is presented in the “Materials and Methods” section, containing a detailed description of the experiment and the logic of its conduct. As there were no signs of cell death and significant cytotoxicity after 24 hours of exposure to the tested compounds, we concluded on the safety of liposomal G-CSF.
Response:
The fact that the cells were first acclimatised to the culture condition before addition of the inhibitors needs to be made clearer. Furthermore, an assay that extends for only 24 hrs might be described as an assay for acute cytotoxicity, at best. This point will be mentioned again later.
The Section 2.5 states: «…the current study focuses on quantification of mitochondrial oxygen consumption rate (OCR). OCR is critically important for cellular metabolism evaluation, associated with mitochondrial respiration and ATP production. OCR measurement provides a comprehensive insight into cellular bioenergetics and allows determining mitochondrial toxicity by measuring mitochondrial stress…».
We assessed the mitochondrial toxicity by analyzing the influence of tested compounds on cellular metabolism after 24 hours.
C22. It is hard to follow their analyses for mitochondrial function. Thank you for your comment. We confirm that a reliable difference is observed at basal respiration stage (4.30±0.07 for controls vs 5.84±0.12 for 10 mg/ml, p = 0.000011) and after oligomycin introduction (2.47±0.09 for controls vs 4.32±0.1 for 10 mg/ml, p = 0.000003), which inhibits ATP synthesis. The absence of reliable difference at subsequent stages, such as maximal respiration and respiratory chain inhibition, indicates a lack of significant proton leakage. This confirms the reversible nature of the effect in the context of cellular respiration leading to a conclusion that there are no significant cell morphological changes.
Response:
This is actually a very detailed explanation that should have been included for easier understanding. However, the data suggests no significant metabolic, not morphological, cellular changes. No evidence on morphological change had been presented at all.
Corrected.
C23. When adding G-CSF-loaded liposomes to the cells, the authors describe addition of test compounds at different concentrations (1.25; 2.5; 5; 10 mg/mL) (Lines 416-417). These concentrations refer to the G-CSF. Comparable concentrations were used for samples containing unencapsulated G-CSF after consideration of the G-CSF encapsulation degree, as well as other factors that may have arisen during G-CSF production, such as protein stability and variability of different batches. This ensured a correct comparison of liposomal and unencapsulated G-CSF for analysis of their effect in the experiment.
Response:
This is an important detail that is missing from the Methods.
This has been included in the "Materials and Methods" section.
C25. What is the significance of the “dose-dependent effect of G-CSF liposomes” (Lines 454-455)?
The concept of "dose-dependent effect of G-CSF liposomes" refers to the changes in cellular response with changing concentrations of G-CSF-loaded liposomes. Our study established a positive correlation between mitochondrial respiration (basal respiration and after oligomycin administration) and increasing concentrations of G-CSF-loaded liposomes. The obtained data are important for selection of optimal dosages and administration regimens, which can be beneficial for the future clinical application of liposomal of G-CSF.
Response:
The reviewer is well-aware of what a dose-dependent response is. What was not clear was the
meaning behind the observation of the dose-dependent response.
Here, the authors explain that this information would guide the selection of dosages and
administration methods. This is another explanation that should have been inlcuded in the report.
The section 2.5 states: "However, the difference in oxygen consumption when administering various concentrations of G-CSF-loaded liposomes warrants further investigation to determine their effective and safe concentration." This explanation reflects our understanding of a dose-dependent response importance. We agree that this could have been discussed in more detail; however, we believe that the named sentence gives an overview of how further studies might influence dosage selection and administration routes.
C26. Were other cellular functions evaluated?
Current study aimed to evaluate the cell metabolic activity based on MTT analysis. However, cell morphology during the experiment was assessed by microscopy.
Response:
No data on the cell morphology in response to exposure to the liposomes was presented. This would have provided important additional, and basic, information on the cytotoxicity of the formulations.
As previously mentioned, the section 2.5 states: «…the current study focuses on quantification of mitochondrial oxygen consumption rate (OCR). OCR is critically important for cellular metabolism evaluation, associated with mitochondrial respiration and ATP production. OCR measurement provides a comprehensive insight into cellular bioenergetics and allows determining mitochon-drial toxicity by measuring mitochondrial stress…».
C27. Did the authors perform simple viability assays which track growth and reproduction? They explained (although not clearly) why the MTT assay was excluded, but what about other enzymebased methods such as calcein-AM staining. Etc?
Our study preliminary assessed the safety of G-CSF-loaded liposomes in human dermal fibroblasts due to their versatility for in vitro safety evaluation. We recognize the importance of viability assays, including calcein-AM staining, and we plan to add this method in future research. In addition, we plan to conduct an extended analysis on different cell types for comprehensive characterization of the safety and efficacy of G-CSF-loaded liposomes.
Response:
Once again, this would have been important additional, and basic, information on the cytotoxicity of the formulations.
As stated earlier, the goal of the present study was an initial evaluation of the cytotoxicity of G-CSF-loaded liposomes with respect to human dermal fibroblasts. We used these cells due to their versatility and suitability for assessing overall safety in vitro.
C28. Did the cells continue to live and grow aner exposure to G-CSF-loaded liposomes?
During the experiments in Seahorse XFe96 Real-Time Cell Metabolic Analyzer, the administration of rotenone and antimycin A resulted in cell death due to mitochondrial respiration inhibition. Nevertheless, under visual control cells remained viable after treatment with GSF-loaded liposomes before the mentioned inhibitors administration.
Response:
The question is whether exposure to the liposomes, not the inhibitors, results in cell harm. If evidence for this is available, it’s strange that such important information was omitted.
If liposomal formulations exerted a critical impact on cells, we would not observe cellular respiration at any stage. However, only G-CSF solution at the highest concentration reliably influenced cellular basal respiratory activity.
Response:
The questions posed from C26 – C34 do not arise from idle curiosity. Answers to them will provide more tangible information about how the liposomes behave toward, or in the presence of, the test samples. This will give clues to whether the test subject would develop adverse reactions after the period of the acute cytotoxicity testing.
Declarations of safety, or lack of toxicity, or biocompatibility, etc, have always been controversial, and for good reason. Researchers are usually careful about making clear the definitions of these terms as well as the limits and caveats of their conclusions.
All of this would not be of such importance, if the authors had not insisted on broadly declaring their liposomes as being ‘safe’. Such a confident conclusion is not justified based only on the presented mitochondrial toxicity data.
To put it crudely, based solely on the data presented, would the authors be confident that a patient would not be harmed when exposed to these liposomes?
Also, the “Conclusion” states: "To fully understand the effectiveness of the obtained liposomes in biomedical applications, further research is needed, including an evaluation of their effectiveness in vivo. Such studies will allow a deeper understanding of the behavior and efficacy of G-CSF-loaded liposomes under controlled conditions and in living systems. This will help better understand their potential in biomedical applications such as the treatment of various diseases, including those where immune system activation and granulocyte stimulation are important factors."
Thus, we do not claim on unconditional safety of liposomes. Such statement requires further in vitro and in vivo experiments.
- Presentation
D1. Many company names in the Materials section are not properly or correctly given. Some company names given in the manuscript are located in Russian Federation, explaining difficulties in perception.
Response:
This is a strange response. When the name of a company is given in writing, the reviewer does not see how perception can be difficult. Any researcher, inside or outside of the Russian Federation, ought to be able to identify and locate a source provided. This is part of the crucial effort at producing quality assurance and reproducibility. Googling “Scientific and Technological Center BioInvest 330 LLC” shows “Bio Invest Scientific Technology Centre Llc” in Moscow, Russian Federation as the closest hit.
Corrected. However, here are some examples:
